# Multi-layered maps of neuropil with segmentation-guided contrastive learning

Sven Dorkenwald[1,2,3,7], Peter H. Li [1,7], Michał Januszewski [4], Daniel R. Berger[5], Jeremy Maitin-Shepard[1], Agnes L. Bodor[6], Forrest Collman[6], Casey M. Schneider-Mizell[6], Nuno Maçarico da Costa [6], Jeff W. Lichtman [5] & Viren Jain [1]✉

Maps of the nervous system that identify individual cells along with their type, subcellular components and connectivity have the potential to elucidate fundamental organizational principles of neural circuits. Nanometer-resolution imaging of brain tissue provides the necessary raw data, but inferring cellular and subcellular annotation layers is challenging. We present segmentation-guided contrastive learning of representations (SegCLR), a self-supervised machine learning technique that produces representations of cells directly from 3D imagery and segmentations. When applied to volumes of human and mouse cortex, SegCLR enables accurate classification of cellular subcompartments and achieves performance equivalent to a supervised approach while requiring 400-fold fewer labeled examples. SegCLR also enables inference of cell types from fragments as small as 10 μm, which enhances the utility of volumes in which many neurites are truncated at boundaries. Finally, SegCLR enables exploration of layer 5 pyramidal cell subtypes and automated large-scale analysis of synaptic partners in mouse visual cortex.

Biological understanding has been enabled by annotating parts of organisms and elucidating their interrelationships. In the brain, numerous types of neuronal and glial cells have been discovered and cataloged according to their morphological, physiological and molecular properties[1–5], typically using methods that interrogate cells in a sparse or isolated setting. Further discoveries would be enabled by producing maps that contain dense assemblies of cells and multiple layers of annotation in the context of a neural circuit or region[6–10].

Producing dense maps of neuropil is challenging due to the multiple scales of brain structures (for example, nanometers for a synapse versus millimeters for an axon)[11], and the vast number of objects in neuropil that must be individually segmented, typed and annotated. Volumetric electron microscopy is an effective way to image brain structures over both large and fine scales[12–14], and automated

segmentation of volume electron microscopy data has also shown substantial progress[15–19], including the demonstration of millimeter-scale error-free run lengths[20].

Automated methods have also been used to infer annotations of cell fragments, and have usually been trained in a supervised fashion for a specific task[9,21–27]. Self-supervised learning has emerged as an alternative to supervised learning, and is capable of producing representations of text[28] and images[29] without the use of labeled data. Specifically, contrastive learning is a commonly used method in which a model is trained to produce similar representations for augmented versions of the same two-dimensional (2D) or three-dimensional (3D) input image. This approach has been used to explore image retrieval in 2D and 3D datasets[30,31], and to analyze coarse neuron morphologies[32], synaptic ultrastructures[33]

[1]Google Research, Mountain View, CA, USA. [2]Princeton Neuroscience Institute, Princeton University, Princeton, NJ, USA. [3]Computer Science Department, Princeton University, Princeton, NJ, USA. [4]Google Research, Zürich, Switzerland. [5]Department of Molecular and Cellular Biology, Center for Brain Science, Harvard, Cambridge, MA, USA. [6]Allen Institute for Brain Science, Seattle, WA, USA. [7]These authors contributed equally: Sven Dorkenwald, Peter H. Li. ✉e-mail: viren@google.com

and morphologies of non-neuronal cells[34] in segmented electron microscopy datasets.

Here, we address the problem of efficiently inferring types and annotations of segmented structures by introducing segmentation-guided contrastive learning of representations (SegCLR), a self-supervised machine learning approach that is scalable in three important respects: first, precomputed SegCLR representations can be used for a diverse set of annotation tasks (for example, identification of subcompartments or cell types); second, SegCLR representations enable accurate downstream analyses with simple linear classifiers or shallow networks; and third, SegCLR representations reduce the amount of ground truth labeling required for specific tasks by orders of magnitude. Perhaps most intriguingly, we show that SegCLR enables a type of annotation that is challenging for either automated methods or human experts: inferring the cell type from a short length (~10–50 μm) of cortical cell fragment. This capability has important implications for the utility of electron microscopy datasets that so far encompass only subsets of whole brains.

SegCLR takes inspiration from advances in self-supervised contrastive learning[29] while introducing a segmentation-guided augmentation and loss function in which positive example pairs are drawn from nearby, but not necessarily overlapping, cutouts of the same segmented cell. In addition to raw volumetric data, this approach therefore also requires 3D segmentation of individual cells throughout the volume, which is a typical requirement for subcellular annotation methods. As we demonstrate, current automated segmentation methods are sufficiently accurate to be used to train SegCLR without further human proofreading. We also show that SegCLR can be combined with Gaussian processes[35] to provide a practical means of uncertainty estimation. Finally, we demonstrate the application of SegCLR to biological inference by detailed cell type analysis of upstream and downstream synaptic partners in mouse visual cortex.

## Results

### Training and inference of SegCLR embeddings

SegCLR enhances the analysis of electron microscopy reconstructions by producing tractable 'embeddings': vector representations that capture rich biological features in a dimensionally reduced space, and in which vector distance maps to a concept of biological distinctness (Fig. 1). These embeddings capture characteristics relevant to a range of downstream tasks without manual feature engineering. Depending on the downstream application, embeddings can also be deployed without any requirement for manual proofreading or ground truth labeling, or with these requirements substantially reduced[36]. Each SegCLR embedding represents a local 3D view of electron microscopy data, and is focused on an individual cell fragment within dense neuropil via an accompanying segmentation. Computed for billions of local views across large connectomic datasets, embeddings can directly support local annotation tasks (Fig. 2), or be flexibly combined at larger scales to support annotation at the level of cells and cell fragments (Figs. 3–5), or circuits (Fig. 6).

SegCLR extends the contrastive learning approach[29,36] by leveraging freely available dense automated instance segmentations of neurons and glia[12,13]. Contrastive methods aim to learn representations by maximizing agreement between matched ('positive') examples in a learned latent space. SegCLR selects example pairs with respect to the segmentation: positive pairs are drawn from nearby locations (along a ≤150 μm skeleton path length) on the same object and trained to have similar representations, while negative pairs are drawn from separate objects and trained to have dissimilar representations (Fig. 1a). We also leveraged the segmentation for input preprocessing: local 3D views of electron microscopy data, 4–5 μm on a side at a voxel resolution of 32–40 nm, were presented to the embedding network after being masked to feature only the segmented object at the center of the field of view (Fig. 1b). The network architecture was based on ResNet-18

(ref. 37), with convolutional filters extended to three dimensions and three bottleneck layers reducing the representation to a 64-dimensional embedding. During training, a projection head further reduced the output to 16 dimensions, on which the contrastive loss was applied[36] (Fig. 1b).

We trained SegCLR separately on two large-scale, largely unproofread, publicly available electron microscopy connectomic datasets, one from human temporal cortex (h01)[13] and one from mouse visual cortex (MICrONS)[12], that were produced via different imaging and segmentation techniques. We then inferred SegCLR embeddings with overlapping fields of view over all non-trivial objects (at least 1,000 voxels) in each volume. This produced a 64-dimensional embedding vector for each masked local 3D view, for a total of 3.9 billion and 4.2 billion embeddings for the human and mouse datasets, respectively. SegCLR thus adds modest storage overhead relative to the full electron microscopy dataset size (human: 980 GB versus 1.4 PB at 4 × 4 × 33 nm; mouse: 1 TB versus 234 TB at 8 × 8 × 40 nm). Visualization of an illustrative subset of the resulting embeddings after dimensionality reduction via UMAP (uniform manifold approximation and projection)[38] showed the structure across each embedding space (Fig. 1c,d). Visualization of embeddings over individual cells also showed the structure within and between them (Fig. 1e,f), suggesting the potential for embeddings to solve diverse downstream tasks.

### Cellular subcompartment classification

Embedding vectors representing local segment views throughout the electron microscopy datasets can be applied to a variety of downstream tasks, such as clustering, similarity search or classification (Fig. 2a). We first examined the use of SegCLR embeddings to distinguish cellular subcompartments such as axons, dendrites and somas (Fig. 2b). In the human cortical dataset we also included astrocytic processes, as a distinct subcompartment for which we had ground truth labeling. On a set of segmented object locations with expert labeled subcompartment identities, the respective SegCLR embeddings formed largely separable clusters in embedding space (Fig. 2c,e). A linear classifier trained to distinguish embeddings from the human cortical dataset could identify subcompartments in a held out test set with a mean F1 score of 0.997, while for the mouse dataset classification the mean F1 score reached 0.958. The F1 score summarizes classification accuracy and reflects both precision and recall performance via their harmonic mean (Supplementary Table 3).

We also tested reducing the ground truth labeling requirements, and compared the performance of subcompartment classification using SegCLR embeddings with that of a directly trained, fully supervised subcompartment classification network[24] at several reductions. The supervised network input data and network architecture (ResNet-18) were identical to the SegCLR setup, except that we replaced the SegCLR bottleneck and contrastive projection head with a classification softmax. On the 4-class h01 subcompartment task, the fully supervised model reached an F1 score of 0.993 when trained on the full dataset ($n$ = 2,846,921). The embedding-based approach exceeded the performance of the fully supervised approach for all sample sizes, and still achieved high performance for small samples where the fully supervised approach performed poorly. We estimated the reduction in required ground truth labels at the point where the performance of the fully supervised model plateaued (10% of all labels; $n$ = 284,692; F1 score = 0.991). Our embedding-based classification matched this performance with approximately 400-fold less labeled training data ($n$ = 695). We also note that for the smallest sample sizes, certain random samples performed much better than the average random sample (Fig. 2d,f, light gray points), suggesting a potential for further gains in accuracy and efficiency under more sophisticated sampling strategies[39].

In the experiments above, both the SegCLR model and the linear classifier were trained on the target dataset. To test whether a SegCLR

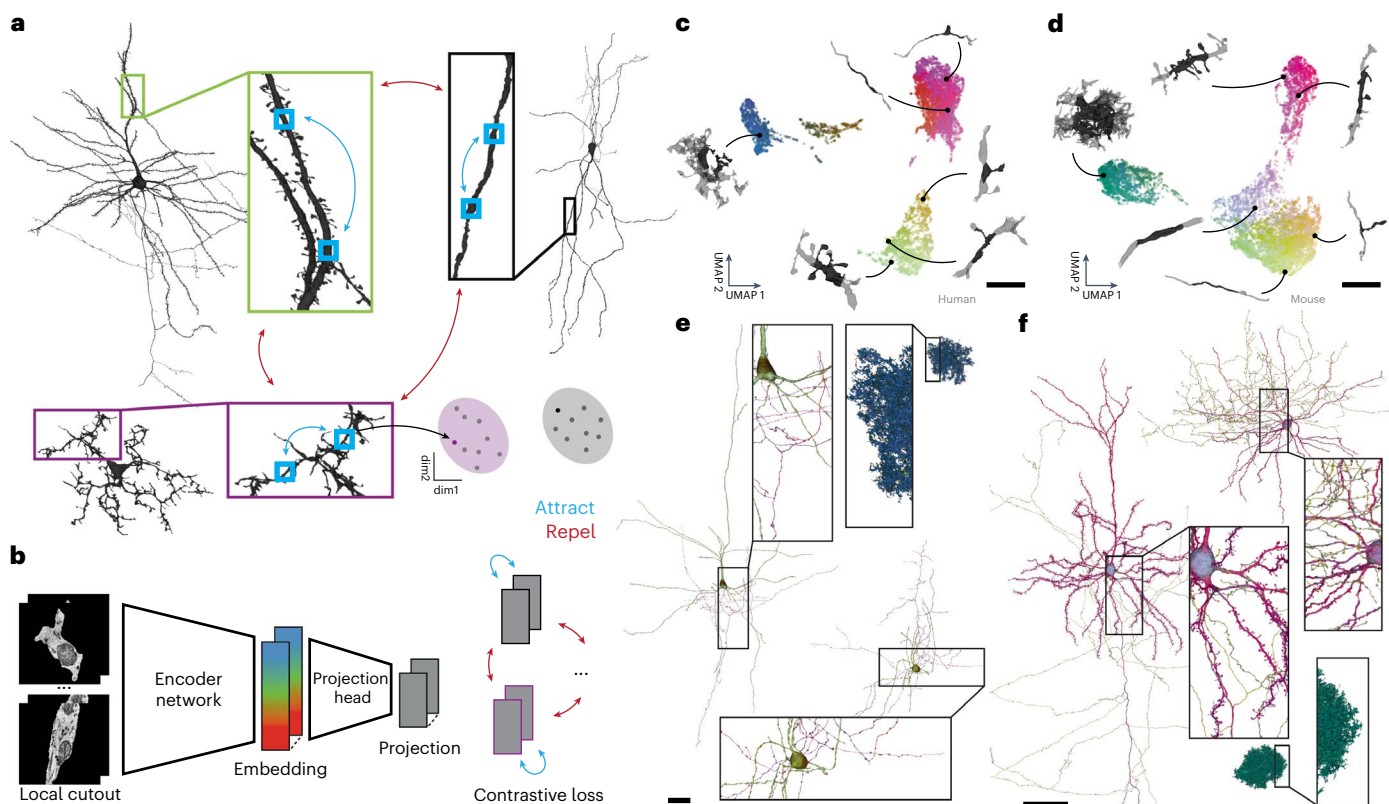

**Fig. 1 | SegCLR. a**, In SegCLR, positive pairs (blue double-headed arrows) are chosen from proximal but not necessarily overlapping 3D views (small blue boxes) of the same segmented cell, while negative pairs (red double-headed arrows) are chosen from different cells. The SegCLR network is trained to produce an embedding vector for each local 3D view such that embeddings are more similar for positive pairs than negative pairs (cartoon of clustered points). **b**, The input to the embedding network is a local 3D view (4.1 × 4.1 × 4.3 µm at 32 × 32 × 33 nm resolution for human data; 4.1 × 4.1 × 5.2 µm at 32 × 32 × 40 nm resolution for mouse) from the electron microscopy volume, masked by the segmentation for the object at the center of the field of view. An encoder network based on a ResNet-18 is trained to produce embeddings, via projection heads and

a contrastive loss that are used only during training. **c,d**, Visualization via UMAP projection of the SegCLR embedding space for the human temporal cortex (**c**) and mouse visual cortex (**d**) datasets. Points for a representative sample of embeddings are shown, colored via 3D UMAP RGB, with the corresponding 3D morphology illustrated for six locations (network segmentation mask input in black, surrounded by 10 × 10 × 10 µm context in gray; masked electron microscopy input data not shown). **e,f**, Embeddings visualized along the extent of representative human (**e**) and mouse (**f**) cells. Each mesh rendering is colored according to the 3D UMAP RGB of the nearest embedding for the surrounding local 3D view. Some axons are cut off to fit. Scale bars: **c,d**, 5 µm; **e,f**, 100 µm.

model trained on a different dataset could still be used to classify subcompartments with high accuracy, we repeated our experiment on the h01 dataset, but used the SegCLR model trained on the MICrONS dataset to produce the embeddings (Extended Data Fig. 1). A linear classifier trained on h01 labels using the MICrONS model embeddings achieved slightly lower performance than when using the h01 model embeddings (mean F1 score, 0.975), but overall accuracy was still high and replicated the good scaling behavior for small sample sizes, where this classifier beat the fully supervised model (<0.1% of all labels, $n < 2,846$, Extended Data Fig. 1d). Thus trained SegCLR models can be transferred to new datasets effectively without retraining, reducing barriers to adoption.

Finally, we asked whether minimal refinement of SegCLR embedding models on a new dataset can improve performance without requiring a large computer cluster. We tested the fine-tuning of the MICrONS embedding model on h01 data using a single TPU (tensor processing unit; TPUv2, 8 GB memory) for training (without freezing any layers). A linear subcompartment classifier trained on these refined embeddings indeed demonstrated improved accuracy (mean F1 score of 0.984 versus a mean F1 score of 0.975 without refinement, 0.997 by the linear classifier using embeddings from a SegCLR model trained on the h01 data, and 0.993 by the fully supervised approach) (Extended Data Fig. 1d).

## Classification of cell types for large and small fragments

Cell type classification is another important application for biological analysis of dense neuropil. To assess performance on this task, we focused on neuron and glia types for which we had expert ground truth labels: 13 mouse cell types and 6 human cell types (Fig. 3a). We also restricted the labeled set to manually proofread cells, to avoid cell type ambiguities from any residual merge errors in the reconstructions. This restriction was important for the evaluation of SegCLR, but does not limit its applicability for segmentation datasets that lack proofreading.

Although individual SegCLR embeddings representing local 3D views 4–5 µm on a side were sufficient for subcompartment classification (Fig. 2), for cell typing we found it helpful to aggregate embedding information over larger spatial extents prior to classification (Fig. 3c). Starting from a position of interest on a cell, we collected all nearby embeddings for a set distance $R$ along the skeleton path in all directions. We then combined the collected set of embeddings by computing a mean embedding value over each feature dimension; this simple aggregation approach proved effective as input to the shallow two-module ResNet classifiers used for cell typing.

Ultimately we wanted to classify not only large but also small cell fragments, which are common in automated reconstructions. Therefore, an important question was how many embeddings, over what spatial extent, need to be aggregated to achieve good cell typing.

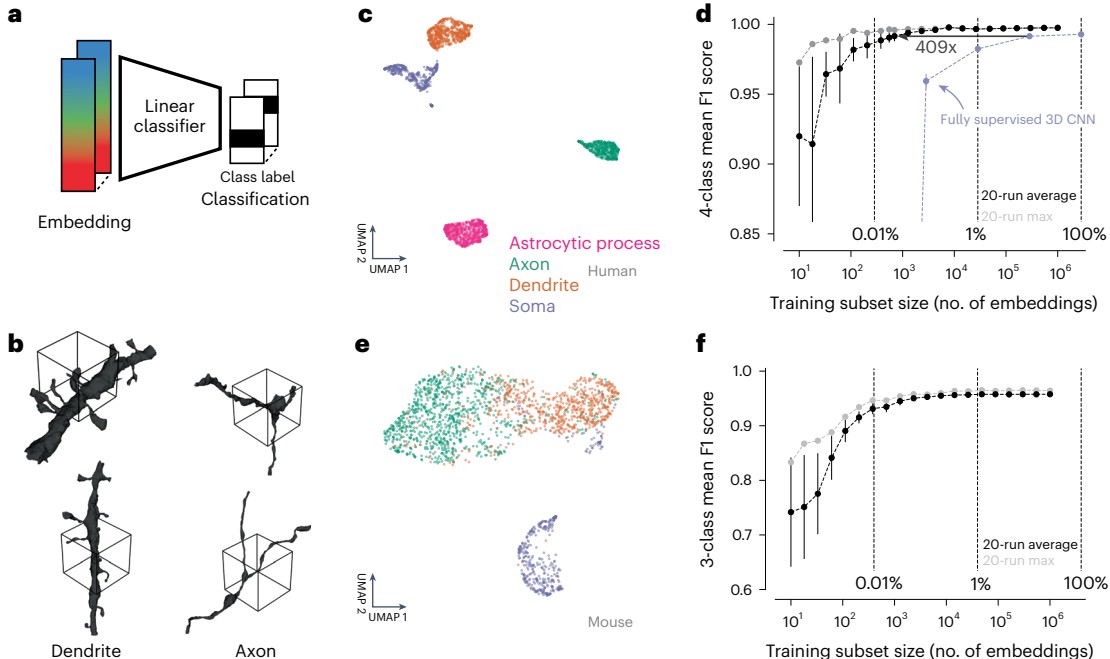

**Fig. 2 | Subcompartment classification of SegCLR embeddings. a**, Embedding vectors computed across the extent of the electron microscopy datasets can be used as compact inputs to downstream tasks, such as subcompartment classification. Each embedding represents a single local view (~4–5 µm on a side). **b**, Ground truth examples of axon and dendrite subcompartment classes from the human temporal cortex dataset. The local 3D views for single embeddings are indicated by the wireframe cubes. The embedding network also receives the electron microscopy image data from within the segment mask. **c**, Embedding clusters from the human cortical dataset visualized via 2D UMAP. Each point is an embedding, colored by its ground truth subcompartment class as judged without reference to the embeddings. **d**, Evaluation of linear classifiers trained for the subcompartment task on the human dataset. The mean F1 score across

classes was computed for networks trained using varying sized subsets of the full available training data. For each training set sample size, mean and standard deviation of multiple subset resamplings are shown (error bars are obscured by the points for larger sample sizes). Light gray points show the best class-wise mean F1 score obtained for any training subset sampled at a given size. The light blue line indicates the performance of a fully supervised ResNet-18 classifier (a convolutional neural network, CNN) trained on the full and subsets of the available training data. Error bars are s.d. (*n* = 20 subsamples). See Supplementary Table 1 for the number of training samples per class. **e**, As in **c**, for the mouse visual cortex dataset and three ground truth classes (axon, dendrite, soma). **f**, As in **d**, for the mouse dataset. Error bars are s.d. (*n* = 20 subsamples). See Supplementary Table 1 for the number of training samples per class.

We assessed this by constructing varying sized cutouts from the ground truth cells (Fig. 3b) corresponding to different aggregation distances *R*. For both mouse and human datasets, SegCLR supports high-accuracy cell typing at aggregation distances of only 10–25 µm (human 6-class mean F1 score of 0.957 at *R* = 10 µm; mouse 6-class mean F1 score of 0.967 at *R* = 25 µm; mouse 10-class mean F1 score of 0.832 at *R* = 25 µm; mouse 13-class mean F1 score of 0.748 at *R* = 25 µm) (Fig. 3d–h). When using all 13 mouse ground truth types, most residual classification errors were between pyramidal cell subtypes (Fig. 3h and Extended Data Fig. 2), particularly in their axons.

**Unsupervised data exploration via SegCLR**
SegCLR also proved useful for data exploration beyond supervised classification tasks. Unsupervised UMAP[38] readily separates putative subcompartment types, when sampling embeddings uniformly (Fig. 1c,d) as well as in manually labeled subsets (Fig. 2c,e). Focusing on subregions of embedding space reveals finer-grained distinctions. For example, embeddings representing dendritic arbors of layer 5 pyramidal cells contain further structure, featuring three main distinct UMAP clusters (Fig. 4a).

Visualizing locations for which the embeddings fell within cluster 1 revealed a strong association with apical dendrite subcompartments (Fig. 4b) across all 181 cells examined, whereas locations outside cluster 1 were primarily basal dendrites. In contrast, we found that cluster 2 comprised basal dendrites restricted to only a small subset (13%) of cells, with distinctive dendritic morphology featuring relatively little

branching and basal dendrites descending laterally over long distances (Fig. 4b, inset).

Cluster 3 contained the basal dendrite nodes for the remaining majority (87%) of layer 5 pyramidal cells. Visualizing their entire reconstructed morphologies showed that a subset of these cells had distinctive axon trajectories, consistent with their axons joining a major output tract (Fig. 4c, left). The dendritic embeddings for these 'tract' cells occupied only half of cluster 3. We then selected 30 cells from cluster 3 that were mostly excluded from the tract subregion. None of these 30 showed the axon tract morphology (Fig. 4c, right). Thus, basal dendrite embeddings can largely separate two pyramidal subgroups, which independently separate based on axon trajectories.

Based on its distinct morphology and relative rarity, the cluster 2 group (Fig. 4b) probably corresponds to the type described as near-projecting (NP) or cortico-cortical non-striatal (CC-NS)[40–42] cells. The cluster 3 tract group (Fig. 4c, left) probably corresponds to the type described as extra-telencephalic (ET), cortico-subcortical (CS), thick-tufted (TT) or pyramidal tract (PT), which provide the primary cortical output onto subcortical brain areas[40–42]. The cluster 3 no-tract cells (Fig. 4c, right) then probably correspond to the intra-telencephalic (IT) or cortico-cortical (CC) type. Although these subtypes have previously been distinguished by gene expression, developmental history, electrophysiology, morphology or their distal projection targets, we add evidence that they can be distinguished by their dendritic morphology and proximal axonal trajectory, within both primary visual area (V1) and higher visual areas (HVA) (Fig. 4c).

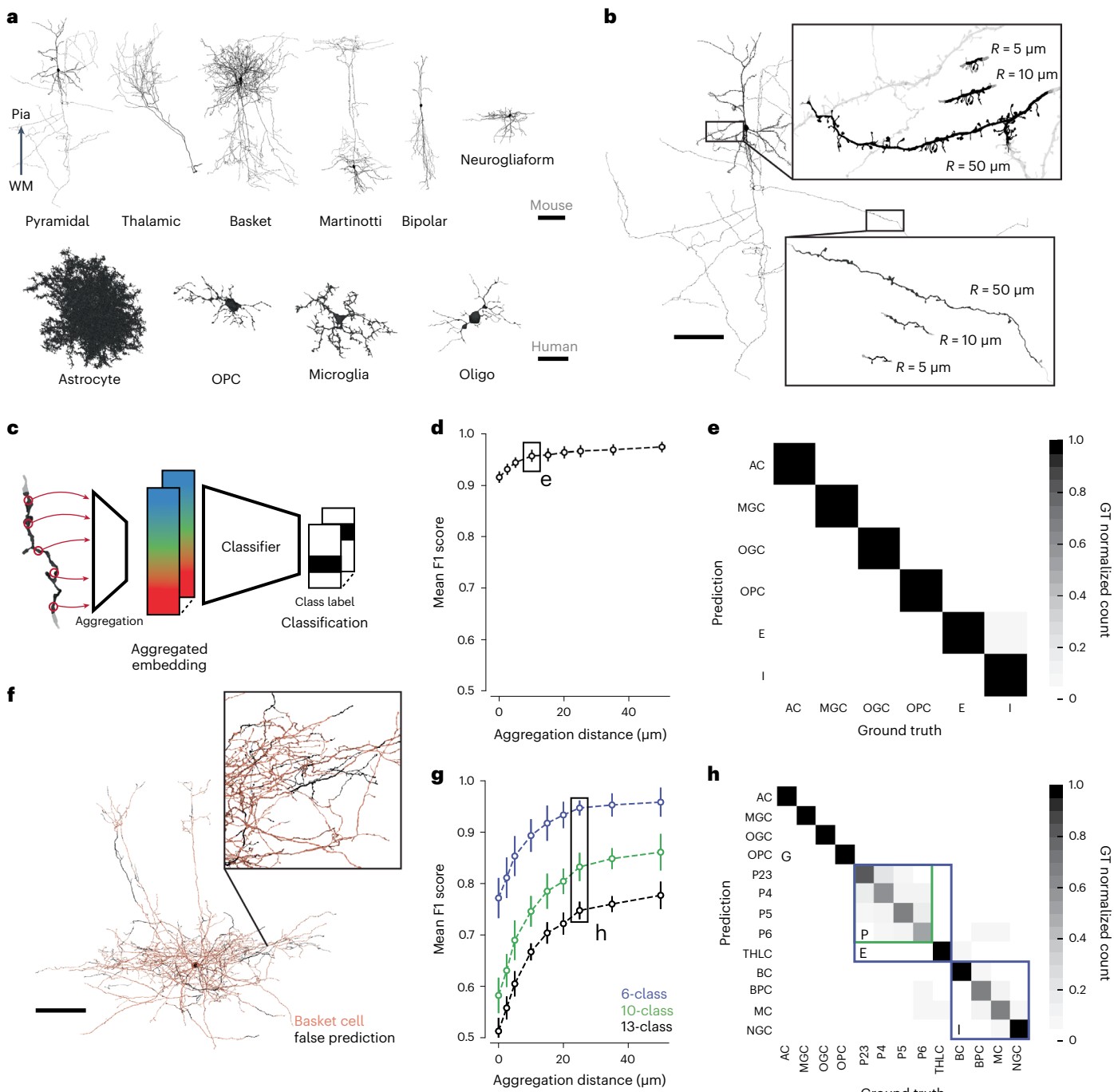

**Fig. 3 | Cell type classification of large and small cell fragments via aggregated embeddings. a**, 3D renderings of representative proofread neuron and glia cells, for a selected subset of the types used in the mouse and human datasets. The pyramidal cell axon is cut off to fit. Cells are oriented from white matter (WM) to pia. **b**, Rendering of representative cutouts from a pyramidal dendrite (top inset) and axon (bottom inset). Different size cutouts are defined by the skeleton node aggregation radius $R$. **c**, Cell type classifiers are trained on top of SegCLR embeddings after aggregation into a mean embedding over the cutout. **d**, Cell typing performance of shallow ResNet classifiers over different aggregation radii for the six labeled cell types in the human dataset. Zero radius corresponds to a single unaggregated embedding node. Error bars are s.d. ($n = 20$ subsamples). See Supplementary Table 2 for the number of training samples per class. **e**, Confusion matrix for the 6-class human cell type task at a 10 μm aggregation radius. GT, ground truth. **f**, Illustration of SegCLR cell type predictions over the extent of a single basket cell from the mouse test set. The orange areas are predicted correctly, while the sparse black areas show

mispredictions. **g**, Cell typing performance for the mouse dataset. The 13-class task (black) uses all of the ground truth-labeled classes, while the 10-class task (green) combines all pyramidal cell labels into a single class. The 6-class task (blue) further reduces the neuronal labels into excitatory and inhibitory groups, comparable to the labels available on the human dataset (**d**). Error bars are s.d. ($n = 20$ subsamples). See Supplementary Table 2 for the number of training samples per class. **h**, Confusion matrix for the mouse 13-class cell type task at a 25 μm aggregation radius. Colored boxes indicate the group of four pyramidal cell types that were collapsed into the 10-class task, and the five excitatory and four inhibitory types collapsed into the 6-class task in **g**. Abbreviations: AC, astrocyte; BC, basket cell; BPC, bipolar cell; E, excitatory neuron; I, inhibitory interneuron; MC, Martinotti cell; MGC, microglia cell; NGC, neurogliaform cell; OGC, oligodendrocyte cell; OPC, oligodendrocyte precursor cell; P2–6, cortical layer 2–6 pyramidal cell; THLC, thalamocortical axon. Scale bars: **a**, neuronal 100 μm, glia 25 μm; **b,f**, 100 μm;.

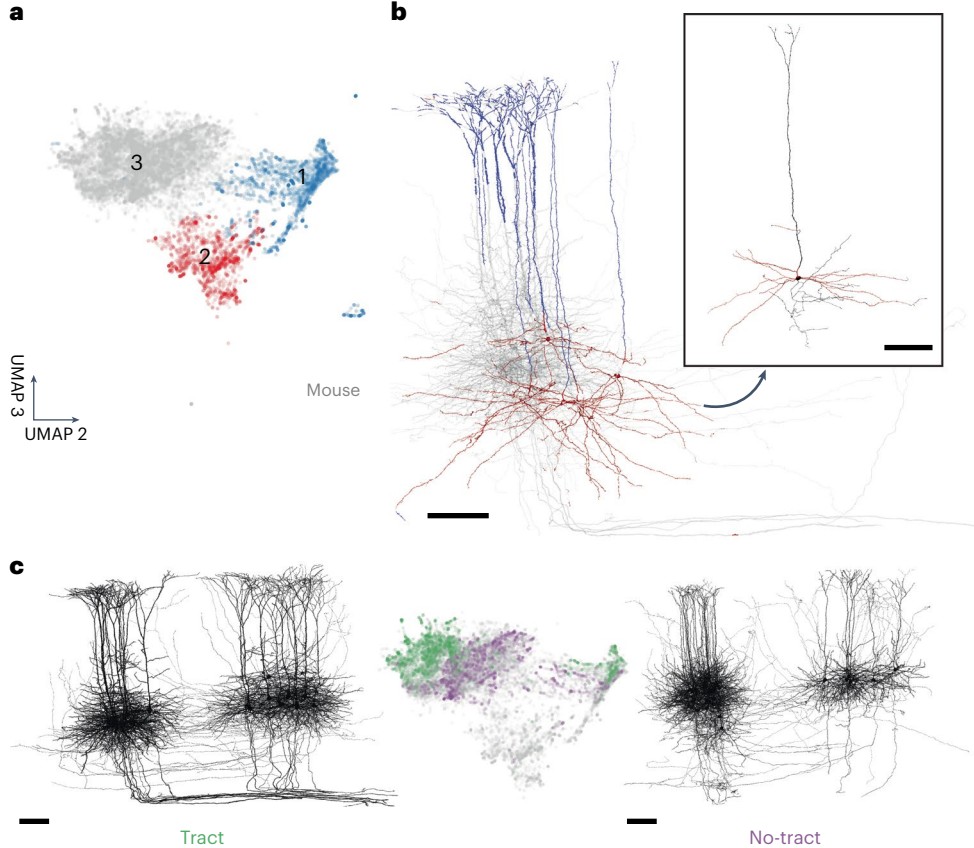

**Fig. 4 | Unsupervised exploration of mouse layer 5 pyramidal dendrite embeddings. a**, SegCLR embeddings projected to 3D UMAP space, with two selected axes displayed. Each point represents an embedding (aggregation distance 50 μm) sampled from only the dendrites of mouse layer 5 pyramidal cells. The UMAP data separate into three main clusters. **b**, Renderings of selected cells, colored to match **a** for locations for which the nearest embedding fell within cluster 1 (blue) or cluster 2 (red). Projections falling within cluster 1 are strongly associated with the apical dendrite subcompartment, while cluster 2 is strongly associated with a subset of basal dendrites corresponding to cells with

a distinct 'near-projecting' (NP) morphology (inset). **c**, Of the cells for which the projections fall within cluster 3, a subset have distinctive axon trajectories consistent with their axons joining a major output tract (left). These 'tract' cells also occupy a distinct subregion within cluster 3 (middle, green). Cells occupying the remainder of cluster 3 (middle, purple) consistently lack the axon tract morphology (right). The tract and no-tract groups are also able to be separated in both primary visual area V1 (left group of cells for both tract and no-tract) and higher visual areas (right group of cells for both tract and no-tract). Scale bars, 100 μm.

## Out-of-distribution input detection via Gaussian processes

A remaining issue with applications such as cell typing on large-scale datasets is how to gracefully handle examples that fall outside the distribution of labeled examples. These 'out-of-distribution' (OOD) input examples could contain imaging artifacts or segmentation merge errors, or they could represent genuine biological structures that were absent in the training set. For example, ground truth labels commonly do not contain all possible cell types, but one wishes to classify the known types across the dataset while avoiding spurious classifications of surrounding segments belonging to diverse unknown types.

We addressed OOD inputs via spectral-normalized neural Gaussian processes[35] (SNGP), which add a prediction uncertainty to the model output (Fig. 5a) based on each example's similarity to the training data. This enables OOD inputs to be detected and rejected, rather than spuriously classified, while requiring no extra labeling effort. To evaluate SNGP, we constructed a human cortical cell type dataset in which only the glial types were used to train classifiers, while both glial and neuronal types were presented for testing (Fig. 5b). The neuronal types, making up 50% of the constructed test set, thus served as an OOD pool.

We first trained a small conventional network (lacking SNGP capabilities) on the glial classification task. Specifically, a shallow two-module ResNet classifier ('ResNet-2') was trained on locally aggregated embeddings (radius 10 μm) from only the labeled glial cells. This network performed with high accuracy on the in-distribution

glial half of the test set, but inherently misclassified all OOD neuronal examples (Fig. 5d). Next, we added an SNGP module to the ResNet-2 (ref. 35), thereby equipping the classifier with an uncertainty output that estimates in part the degree to which each example is OOD with respect to the training distribution (Fig. 5e). This uncertainty can be thresholded at a task-appropriate level to determine how aggressively to reject OOD inputs.

The SNGP-ResNet-2 retained strong in-distribution glia classification performance while effectively filtering out OOD neuronal examples (Fig. 5f). With uncertainty thresholded at a level that optimized F1 on a validation set, and the resulting OOD examples treated as a separate class, the overall mean F1 score reached 0.875 (5-class average). Note that the network layer substitutions for SNGP apply only to the small classifier network, with no modifications required to the underlying SegCLR embeddings. Furthermore, the neuronal ground truth labels used here were needed only to validate the results, while the training and deployment of a classifier with SNGP OOD detection requires no extra ground truth labeling beyond the in-distribution set.

Finally, we also evaluated the spatial distribution of local uncertainty over larger segments. This is particularly relevant for unproofread segments that contain reconstruction merge errors between a labeled and an OOD type. For example, the uncertainty of our SNGP classifier can distinguish neuronal fragments erroneously merged onto a central glia (Fig. 5g). Automated merge error correction based

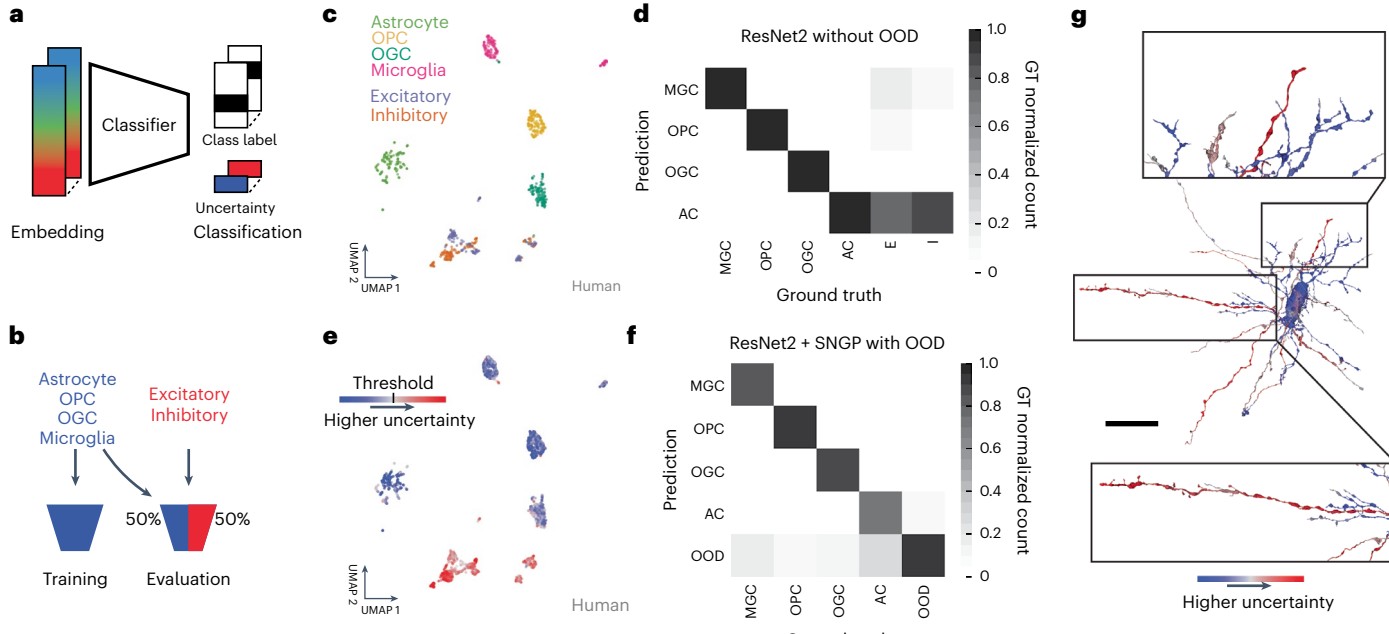

**Fig. 5 | OOD input detection via Gaussian processes. a**, We handled OOD inputs by computing prediction uncertainties alongside class labels, and calibrated the uncertainties to reflect the distance between each test example and the training distribution. **b**, To evaluate OOD detection, we trained classifiers on glial cell type labels, and then evaluated the classifiers on a 50–50 split between glial and OOD neuronal cell types. **c**, UMAP of locally aggregated embeddings (radius 10 μm) from the human cortical dataset, colored by ground truth-labeled cell type. **d**, Confusion matrix for a ResNet-2 classifier trained on only the four glia types, with OOD neuronal examples mixed in at test time. **e**, As in **c** with the UMAP embeddings now colored by their SNGP uncertainty. The colormap

transitions from blue to red at the threshold level used to reject OOD examples in our experiment. **f**, Confusion matrix for the SNGP-ResNet-2, assembled from 20-fold cross-validations. Examples that exceed the uncertainty threshold are now treated as their own OOD predicted class. **g**, Spatial distribution of local uncertainty over an unproofread segment that suffers from reconstruction merge errors between the central OPC glia and several neuronal fragments. The uncertainty signal distinguishes the merged neurites (red, high uncertainty or OOD) from the glia cell (blue, low uncertainty) with a spatial resolution of approximately the embedding aggregation distance. Scale bar, 25 μm. See Fig. 3 for cell type abbreviations.

---

on these branch distinctions[13,24] combined with direct detection of merge-specific embedding features[43] would be an attractive application for future investigation.

## SegCLR cell typing of pre- and post-synaptic partners

In brain circuit analysis, a common goal is to identify the cell types of the thousands of synaptic partners upstream or downstream of a particular cell or circuit of interest[44–46]. Due to the incompleteness of current automated reconstructions, contemporary connectomics-based circuit analysis efforts typically require substantial manual tracing to extend each partner neurite back from the synapse to provide sufficient morphological structure to enable an expert to propose a cell type identity. Furthermore, in many datasets the partner neurites may terminate at a volume boundary before their cell type is manually resolvable. However, as demonstrated here (Fig. 3), SegCLR is able to classify cell types of even relatively small cell fragments with high accuracy, enabling large-scale analysis of synaptic partners without manual tracing.

We leveraged this capability to perform detailed analysis of synaptic partners across multiple cell types in mouse cortex (Fig. 6). For each synapse annotated in this dataset[12] we gathered the predicted cell type annotation of the closest embedding nodes pre- and post-synaptically. We filtered out 9.2% of presynaptic and 7.8% of postsynaptic fragments that were too short ($R_{max} < 2.5$ μm, see Methods) and ignored a further 8.8% and 2.5%, respectively, because of high uncertainty scores.

First, we analyzed the distribution of presynaptic cell types for a representative sample of 919,500 synapses at varying cortical depths (Fig. 6a). Our method classified 72.3% of presynapses and 76.3% postsynapses as excitatory (ignoring those marked as uncertain). We then focused on a core set of proofread cells (Fig. 6b) for which we collected

all their input and output synapses, along with the corresponding (unproofread) presynaptic and postsynaptic partner fragments. For each proofread cell, we then analyzed the relative proportions of received inputs by cell type (Fig. 6c–e) as well as the proportion of downstream targets (Fig. 6h–l).

For upstream (presynaptic) partners, we found differences in the proportion of intracortical (from pyramidal cells) and subcortical (from putative thalamocortical axons) excitatory inputs (Fig. 6e–g). In layer 4, the canonical cortical input layer, pyramidal cells had more subcortical innervation, with the putative thalamocortical synapses contributing 18% of their excitatory inputs (Fig. 6e,f). This percentage agrees with estimates based on immunohistochemistry in mouse V1 (ref. 47) and other mouse cortical regions[48]. When plotting the prevalence of layer 4 thalamocortical synapses along an axis from V1 to HVA, however, we observed a drop in thalamocortical innervation that coincided with the boundary between the regions (Fig. 6g). Thalamocortical projections to HVA had been identified previously[49–51] but had not been quantified, demonstrating how the proposed computational approach can provide quantitative insights into cortical cell type connectivity.

For downstream partners (Fig. 6h–l), we analyzed the relative proportion of output synapses onto excitatory versus inhibitory partners as a function of distance along the presynaptic axon (Fig. 6j–l and Extended Data Fig. 3c,d). For all cortical layers, the distribution of output synapses along pyramidal cell axons was biased towards more inhibitory postsynaptic partners in the proximal regions[52,53], with excitatory downstream partners growing more common with increasing distance from the soma[54]. Our analysis was thus broadly consistent with previous reports for layers 2 and 3 visual[52,53] and entorhinal

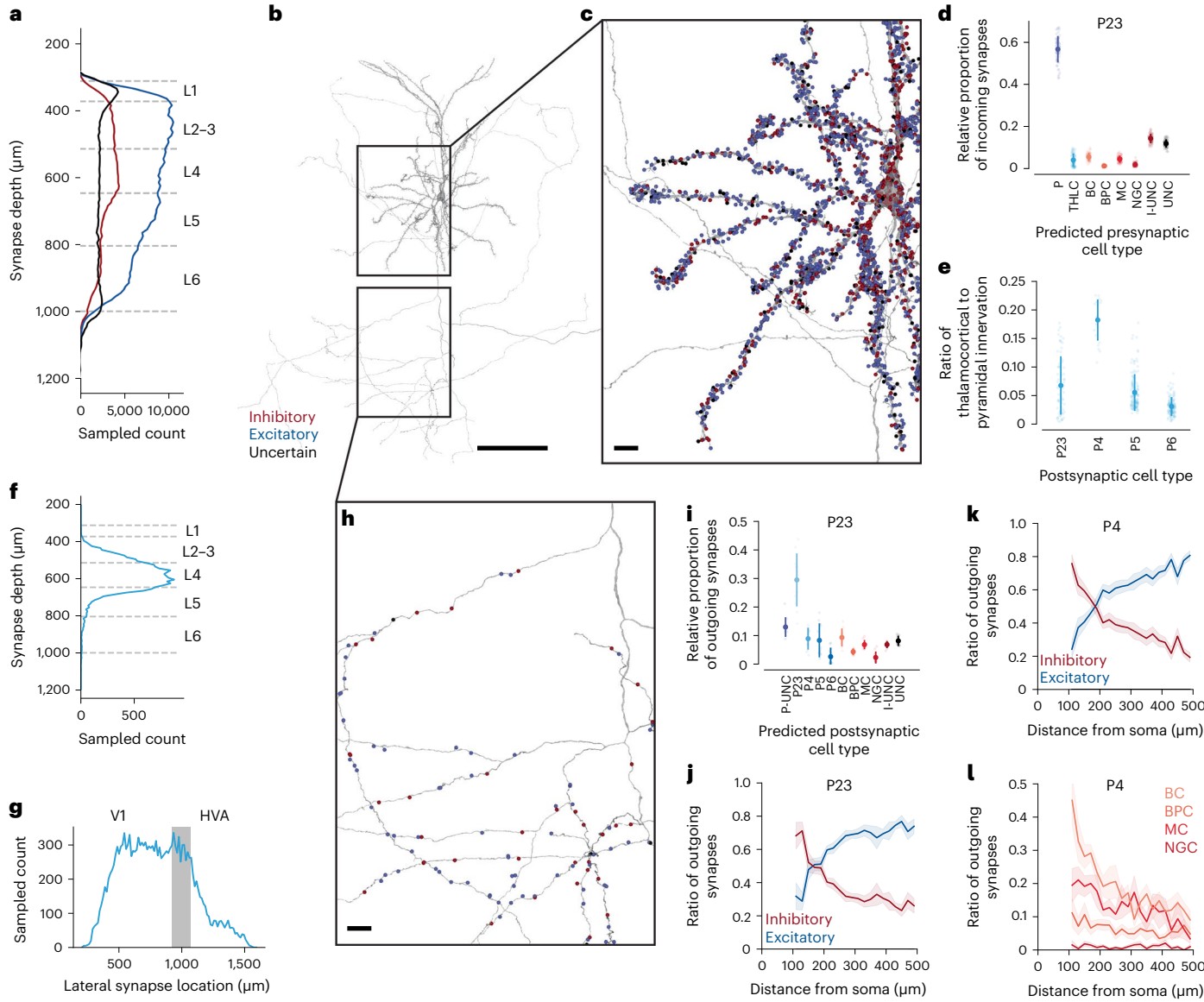

**Fig. 6 | Quantitative analysis of pre- and post-synaptic partner cell type frequencies. a**, The distribution of synapses by predicted inhibitory versus excitatory presynaptic axonal cell type, over the depth of the mouse visual cortex. Lines L1–6 mark the boundaries of the six cortical layers. **b**, Representative layer 2–3 pyramidal cell. **c**, Pyramidal cell input synapse locations, annotated (dots) and colored by predicted inhibitory versus excitatory presynaptic partner cell type. **d**, Distribution of upstream presynaptic partners for layer 2–3 pyramidal cells in V1 with proofread dendrites ($n = 69$). Each cell is represented by a set of points describing its ratios of input cell types. The mean and standard deviation of the ratios are also shown (darker point and line). UNC refers to synapses for which classification was uncertain. I-UNC refers to synapses for which only a coarse classification (excitatory versus inhibitory) could be made. Error bars are s.d. **e**, The ratio of thalamocortical to pyramidal innervation onto pyramidal targets in different cortical layers shows major thalamic input to

cortical layer 4 (P23, $n = 69$; P4, $n = 23$; P5, $n = 136$; P6, $n = 127$). Error bars are s.d. **f**, The thalamocortical synapse counts over the cortical depth. The thalamocortical synapse counts over the cortical depth. Lines L1–6 as in **a**. **g**, The thalamocortical synapse count along the lateral axis through the dataset drops at the boundary between V1 and the HVA. The shaded area shows the approximate projection of the V1–HVA border onto the lateral axis. **h**, As in **c**, for downstream postsynaptic partners. **i**, As in **d**, distribution of downstream postsynaptic partners for layer 2–3 pyramidal cells in V1 with proofread axons ($n = 12$). P-UNC refers to synapses where SegCLR was not able to classify a pyramidal subtype with sufficient certainty. Error bars are s.d. **j,k**, Inhibitory versus excitatory balance of downstream postsynaptic partners with increasing distance along P23 axons ($n = 12$) (**j**) and P4 axons ($n = 19$) (**k**). Mean and s.e.m. are shown for each distance bucket. **l**, As in **k**, divided into individual inhibitory subtypes ($n = 19$). I, inhibitory; P, pyramidal; UNC, uncertain classification. Scale bars: **b**, 100 μm; **c,h**, 10 μm.

cortex[54], but it further showed details of axonal sorting in layer 4 and demonstrated that changes in inhibitory targeting over axonal distance are driven primarily by declines in targeting of basket cells and Martinotti cells.

Using SegCLR cell type predictions for pre- and post-synaptic fragments, we thus derived results consistent with prior reports that relied on laborious manual annotation, while extending their scale in terms of number of synapses, cell types and brain areas analyzed.

## Discussion

We have introduced SegCLR, a self-supervised method for training rich representations of local cellular morphology and ultrastructure, and demonstrated its utility for biological annotation in human and mouse cortical volumes. Beyond the requirement for an accompanying instance segmentation, the current SegCLR formulation has some limitations. First, the 32–40 nm voxel resolution of input views impedes capture of finer electron microscopy ultrastructural features, such as

vesicle subtypes or ciliary microtubule structure. This is a trade-off, because using higher resolution data reduces the field of view of the network (in nanometers) that can be held in accelerator memory and trained effectively. Training SegCLR on higher resolution inputs, or with multiscale capabilities, is worth detailed exploration.

Another limitation is that explicit input masking excludes electron microscopy context outside the current segment, while in some cases retaining surrounding context could be useful, for example for myelin sheaths, synaptic clefts and synaptic partners. We therefore tested a version of SegCLR that receives the unmasked electron microscopy block and the segmentation mask as two separate input channels rather than as a single explicitly masked electron microscopy input. This variant performed similarly on the subcompartment classification task but appeared more sensitive to subtle non-linear photometric differences across the extent of the dataset.

Finally, in the current work we have demonstrated that a simple mean embedding aggregation strategy is sufficient for reasoning over larger contexts. However, more sophisticated aggregation methods[55,56] could still prove useful for generating representations of larger contexts. There are also opportunities to extend representations beyond single cells. For example, neuronal embeddings could be extended with additional dimensions aggregated from pre- and post-synaptic partners to create connectivity-enhanced cell type signatures or to form motif representations.

By providing rich and tractable representations of electron microscopy data, SegCLR simplifies and democratizes downstream research and analysis. The previous state of the art in subcompartment classification required millions of training examples assembled from thousands of manually validated segments, thousands of GPU (graphics processing unit) hours to train a deep network, and hundreds of thousands of CPU (central processing unit) hours to evaluate a large-scale dataset[13,24]. With SegCLR embeddings, this benchmark is outperformed by a linear classifier, trained in minutes on a single CPU, with a few hundred manually labeled examples. We make Python code available for input preprocessing, network training and evaluation, and we release pretrained TensorFlow network weights and training data.

Arguably the most powerful application of SegCLR demonstrated here is the ability to classify neuronal and glial subtypes even from small fragments of cells. This capability has important ramifications, particularly for datasets in which reconstructed cells are incomplete, or in which only a portion of the brain tissue was imaged; identifying connectivity patterns between specific cell types is fundamental to interpreting large-scale connectomic reconstructions.

We release the full SegCLR embedding datasets for the human and mouse cortical volumes to the community, to enhance exploration and understanding of these rich and complex resources.

## Online content

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

## Methods

### Datasets

We used two large-scale serial-section electron microscopy connectomic datasets for SegCLR experiments: one from the human temporal cortex (h01), imaged using scanning electron microscopy[13]; and one from the mouse visual cortex (MICrONS), imaged using transmission electron microscopy[12]. Both datasets are freely available and provide an aligned and registered electron microscopy volume with an accompanying automated dense instance segmentation. For SegCLR experiments we downsampled the human and mouse data to $32 \times 32 \times 33$ nm and $32 \times 32 \times 40$ nm nominal resolution, respectively. The human electron microscopy volume was also normalized using CLAHE (contrast limited adaptive histogram equalization)[57]. We skeletonized both segmentation volumes as previously described[13,26,27,58].

For subcompartment classification (Fig. 2), ground truth human data were collected on an earlier pre-release version of the dataset, as well as two smaller cutouts[13]. All three regions are contained in the publicly released dataset, although they differ slightly in their alignment and photometric normalization. For cell type classification (Fig. 3) it was important for evaluation purposes to have cells with minimal reconstruction merge errors in the ground truth-labeled set. We therefore proofread some cells in the human dataset to exclude regions close to observed merge errors from the embeddings cell type analysis (Fig. 3). On the mouse dataset, we restricted analysis to the set of ground truth-labeled cells that were already expert proofread prior to release[12]. Specifically, we trained SegCLR models on the public segmentation version 117 and then upgraded to the public version 343 for all evaluations and analyses.

### Training SegCLR embedding networks

SegCLR was inspired by SimCLR[29,36]. The embedding network was a ResNet-18 architecture[37] implemented in TensorFlow, with convolutions extended to three dimensions and three bottleneck layers prior to a 64-dimensional embedding output. During training we added three projection layers prior to a normalized temperature-scaled cross-entropy ('NT-Xent') loss[29] with temperature 0.1. The total number of model parameters was 33,737,824.

We also found that downstream task performance was enhanced by adding a decorrelation term to the loss, defined as:

$$L_{\text{decorrelation}} = \frac{1}{d^2 - d} \sum_{i=1}^{d} \sum_{j \neq i} C_{ij}^2 \qquad (1)$$

where $d$ is the embedding dimensionality and $C$ is the correlation matrix between embeddings over the batch. We trained SegCLR networks on $8 \times 8$ v2 Cloud TPUs for up to 350,000 steps with a full batch size of 512 positive pairs (1,024 individual examples) and a learning rate decay schedule starting at 0.2. Separate networks were trained for the human and mouse datasets. Training took approximately 1 week and proceeded at a speed of 0.5–0.6 steps per second.

The input to the network was a local 3D cutout of electron microscopy data 129 voxels on a side, nominally corresponding to $4,128 \times 4,128 \times 4,257$ nm in the human electron microscopy dataset, and $4,128 \times 4,128 \times 5,160$ nm in the mouse dataset. We then masked the electron microscopy data by the segmentation for the object at the center of the field of view, so that possible confounds in the surrounding electron microscopy context would be excluded.

We also leveraged the segmentation and corresponding skeletonization to generate example pairs for contrastive training. For an arbitrary segment, we picked a 3D view to be centered on an arbitrary skeleton node, and then picked a positive pair location centered on a second node on a path length ≤150 μm away on the same skeleton. Positive pairs were preprocessed before training for higher performance. Given that there are more possible pairs for larger distances, we sorted these positive pairs into four distance buckets from which we

drew uniformly. The bucket boundaries were 0, 2,500, 10,000, 30,000 and 150,000 nm.

As in SimCLR[29], we used the 1,022 examples from the rest of the batch, which were drawn from 511 other segments throughout the volume, as negative pairs. We also applied random reflections and photometric augmentations[23] to the inputs, to prevent the network from solving the contrastive task via trivial cues such as the orientation of processes, or the local voxel statistics.

The size of the bottleneck layer is a hyperparameter that determines the embedding dimension (64 in our case). Larger embedding dimensions generally improved performance slightly but increased storage requirements proportional to the embedding dimension and slowed down the downstream analyses accordingly.

### SegCLR model refinement

We refer to refinement or fine-tuning as the further training of a pretrained model on a new dataset. In the present case (Extended Data Fig. 1), we used the model pretrained on the MICrONS dataset and added additional training on the h01 dataset. Specifically, we trained the model with a batch size of 8 positive pairs to test refinement with minimal resources. We used a single TPUv2 (8 GB memory). For context, a P100 GPU commercially released 7 years ago can fit around 12 positive pairs. We trained the model for another 250,000 steps, which took approximately 2 days. We did not freeze any layers of the network and used the same learning rate (scaled to the batch size) as during the original training.

### SegCLR inference

We inferred SegCLR embeddings over the full extent of the human and mouse datasets. After removing trivial segments, we extracted local 3D views centered on skeleton nodes for the remaining segments, with approximately 1,500 nm path length spacing. The set of views for each segment therefore had substantial overlap of approximately 65–70% with typical nearest neighbor views. We then ran SegCLR on all selected views (3.9 billion and 4.2 billion for the human and mouse datasets, respectively) via an Apache Beam Python pipeline running on a large CPU cluster, and stored the resulting embedding vectors keyed by segment ID and spatial XYZ coordinates. Inference proceeded at approximately 1 example per second per CPU with a batch size of 1. For visualization, we ran UMAP dimensionality reduction[38] to 2–4 dimensions on representative samplings or on subsets of interest from among the embeddings. When sampling from a large population of embeddings of local cutouts, we sampled such that all classes were represented with a significant number of examples.

### Subcompartment classification

We trained linear classifiers to identify subcompartment types from embedding inputs based on expert ground truth labels on each dataset (Supplementary Table 1). For comparison, we also trained a fully supervised subcompartment classifier directly on voxel inputs using an identical 3D ResNet-18 architecture and input configuration (photometric augmentation was omitted and 90° 3D rotations were added), and replaced the bottleneck, projection heads and contrastive loss with a classification softmax and cross-entropy loss. The supervised network was trained on 8 GPUs with a total batch size of 64 via stochastic gradient descent with a learning rate of 0.003. Examples were rebalanced class-wise by upsampling all classes to match the most numerous class. We also trained fully supervised classifiers with photometric augmentation added and 90° rotations omitted, to match the SegCLR embedding augmentations, but did not find that this improved performance.

We also trained both SegCLR linear classifiers and fully supervised deep networks on multiple subsamplings to test performance in lower-label regimes (Fig. 2). During subsampling we ensured that every class was represented with at least 10% of the examples and at least three examples in total. For SegCLR linear classifiers we repeated each

sampling round 20 times. For fully supervised classifiers we repeated each sampling 2–5 times for sample sizes <20,000. The fully supervised deep networks were trained for 1.5 million steps, or for 150,000 steps for sample sizes <1,000. We confirmed by inspection of online evaluations that earlier stopping would not have significantly enhanced average performance across classes.

## Cell type classification

We tested cell type classification using a set of expert ground truth-labeled neurons and glia from both human and mouse (Supplementary Table 2). These ground truth cells are generally large and contain somas within the volume but they are not necessarily completely reconstructed. We manually proofread human ground truth cells for merge errors by marking bad agglomeration edges in the agglomeration graph prior to evaluation. In the mouse dataset we restricted analysis to proofread neurons included in the public v343 release. We trained shallow two-module ResNets to predict cell subtypes (Figs. 3, 5 and 6, and Extended Data Fig. 3).

For cell typing, we aggregated local embeddings by collecting all of the embedding nodes in a 0–50 μm radius window along the cell's skeleton path length. A simple aggregation method of taking the mean embedding value over each feature dimension performed well when using shallow ResNets for the downstream task (Figs. 3, 5 and 6) or for unsupervised data exploration (Fig. 4).

We evaluated classification performances by randomly selecting 75% of cells for each cell type for training and 25% for testing. We repeated this sampling 10 times for each aggregation distance and computed the mean across all runs. In each run we computed the F1 score of each class individually and then combined them for an overall mean F1 score. In each run we sampled 30,000 (mouse) or 100,000 (human) embeddings from the cells in the training set, equally distributed across all cell types. During testing, imbalances between classes were balanced by repeating examples from minority classes. The confusion matrices in Fig. 3 were generated by concatenating the test sets of all runs of one aggregation distance first. We then sampled 100,000 test examples equally distributed across all cell types. For the mouse dataset we then repeated this procedure but restricted the sampling to nodes that were automatically labeled as dendrite or axon (Extended Data Fig. 2).

To obtain results for 10 and 6 classes (Fig. 3g) we summed the probabilities of the combined classes. For instance, we summed the probabilities for all pyramidal subtypes to a single pyramidal class for the 10-class evaluation. We then followed the same procedure as for the 13-class evaluation.

## Cell-type ground truth

The different neuron types in the mouse dataset were manually classified based on the morphological and synaptic criteria as described[10]. Pyramidal cells were identified by the presence of a spiny apical dendrite radiating toward the pia, spiny basal dendrites and an axon that formed asymmetric synapses. Putative thalamic axons also formed asymmetric synapses and although their soma was located outside the reconstructed volume, their gross morphology resembles previously described thalamic arbors[59] and their fine morphology is also similar at the ultrastructure level[48]. Neuronal cells were classified as inhibitory interneurons if their axon formed symmetric synapses. Furthermore, the inhibitory interneurons were assigned subtypes using their synaptic connectivity and the morphology of axons and dendrites[60–62]. Basket cells were identified as having a larger number of primary dendrites and as having at least 12% of their postsynaptic targets be pyramidal cell somata. Martinotti cells were identified by having an apical axon that projected to cortical layer 1 and which targeted mostly distal dendritic shafts and spines of excitatory cells, consistent with ref. 60. Martinotti cells were also characterized by having a multipolar dendritic arbor that was usually spiny. Bipolar

cells usually had two to three primary dendrites. The dendrites were usually spiny and showed a vertical bias. Neurogliaform cells were often (but not exclusively) in cortical layer 1. The axons of neurogliaform cells usually have a lower density of synapses, consistent with ref. 60. Neurogliaform cells also had a large number of primary dendrites and have a different pattern of synaptic inputs when compared with other inhibitory cell types.

## Unsupervised exploration of mouse visual cortex layer 5 pyramidal cells

For the detailed exploration of mouse layer 5 pyramidal cells (Fig. 4) we selected only the embedding nodes that were subcompartment classified as dendrites (Fig. 2) from a set of cells that were dendrite proofread and labeled as layer 5 pyramidal cells by human experts ($n = 181$). We then randomly sampled up to 1,051 embeddings per cell for a total of 146,607 and ran unsupervised UMAP[38] to project the embeddings to three dimensions using Manhattan distance, 100 neighbors, 500 epochs and a learning rate of 0.5.

To semi-automatically collect the projections corresponding to the visualized 3D UMAP clusters (Fig. 4a), we ran k-means++ with 25 groups and manually identified which k-means groups corresponded to UMAP clusters 1, 2 and 3. We found that running k-means on the raw embedding space was influenced by spurious dimensions, such as dimensions that primarily picked up subtle differences in the image statistics of groups of z-sections that were captured on different microscopes[12]. By contrast, we found that running k-means directly on 3D UMAP struggled to capture the visual clusters due to their somewhat irregular shapes. Therefore, using k-means on a rerun 5D UMAP, with the same 146,607 input embeddings and with other parameters unchanged, was found to be the most effective.

Selecting cells that had more than 40% of their dendrite projections in cluster 2 ($n = 24$) isolated the near-projecting pyramidal subtype (Fig. 4b, inset). To isolate the 'tract' (putative extra-telencephalic) subset, we rendered all of the cells with clear cluster 3 occupancy, equivalent to those with less than 25% of their projections in cluster 2 ($n = 157$), and then manually selected the subset with distinct output tract axon trajectories ($n = 19$; Fig. 4c, left). This group probably misses some extra-telencephalic cells in our set due to incomplete axon reconstructions, but it clearly identifies a 'tract' subregion of cluster 3 (Fig. 4c, middle) and corresponding 'tract' k-means groups. Finally, to isolate the no-tract (putative intra-telencephalic) subset, we scored all cluster 3 cells based on relative occupancy of 'tract' k-means clusters, and selected a subset ($n = 30$) that was least 'tract' weighted. Visualizing these cells showed that none of them had the tract axon morphology (Fig. 4c, right), providing an inverse confirmation of the clustering.

## Out-of-distribution input detection

We detected OOD inputs via SNGPs[35]. As a baseline we trained a shallow ResNet-2 (two ResNet modules) to classify the glial cell types of 10 μm radius fragments. We then modified the ResNet-2 by spectrally normalizing its hidden layers and replacing the output layer with a Gaussian process, using the SNGP package in TensorFlow. We used the BERT (bidirectional encoder representations from transformers) hyperparameter setting from Liu et al.[35] for our analysis.

We evaluated performance on a test set constructed of a 50–50 split between glia and OOD neuronal examples. First, we computed prediction uncertainty estimates for the test networks using the Dempster–Shafer metric[35]:

$$u(\mathbf{x}_i) = \frac{K}{K + \sum_{k=1}^{K} \exp(h_k(\mathbf{x}_i))} \qquad (2)$$

where $K$ is the number of classes (in our case: 4) and $h_k$ are the classification logits (network outputs prior to probability normalization). As

suggested in the SNGP tutorial, we replaced the Monte Carlo estimation of the output logits with the mean-field method[63]:

$$h_\lambda(\mathbf{x}_i) = \frac{h(\mathbf{x}_i)}{\sqrt{1 + \lambda\sigma^2(\mathbf{x})}} \qquad (3)$$

using $\lambda = 3/\pi^2$ and variances estimated with the Gaussian processes module.

For the evaluation, we repeated the fivefold cross-validation as outlined for the cell type fragment classification. We sampled 100,000 unique examples and upsampled them such that each glia class accounted for 12.5% and each OOD class (excitatory, inhibitory) accounted for 25%. During each fold we trained a new classifier on a subset of the glia examples and then predicted the examples in the hold-out test set as well as the set of neuronal fragments that we reused for all five folds. For each fold we found the uncertainty threshold that maximized the F1 score of the in-distribution versus out-of-distribution task. For this, we set aside half the examples from the test set, which were then not used for calculating the scores. For each fold we replaced the original class prediction (one of four glia classes) with the OOD class when the uncertainty for an example exceeded this threshold. Finally, we calculated F1 scores for each of the five classes after scaling them such that OOD examples accounted for 50% and averaged them for a final F1 score per fold. We reported the mean F1 score across the five folds.

**Automated analysis of synaptic partners**
We applied the best-performing 25 μm ResNet-2+SNGP model from our 13-class cell type classification (Fig. 3) (as well as the same model restricted to three classes) to all embeddings in the dataset. Although classifications from the 13-class model can be aggregated to a 3-class classification, we were interested in the uncertainties produced by the 3-class model to filter out uncertain examples. Here, we set $\lambda = 2$ to compute adjusted logits (eq. 3).

First, we labeled nodes as uncertain when the furthest aggregation distance was below 2.5 μm, indicating that the segment was very small (Extended Data Fig. 3a,b), often containing only a single embedding node. We note that this affects more nodes close to the dataset boundaries where data quality is lower (Fig. 6a). Next, we labeled nodes as uncertain when the predicted uncertainty (eq. 2) was above 0.45 based on manual evaluation of several synapses. For the remaining nodes we assigned the label with the highest classifier probability. Some nodes were predicted to belong to the glia class; for the analysis in Fig. 6 we assigned these to the uncertain category.

Next, we attempted to assign subtype labels to nodes classified as inhibitory or excitatory. For inhibitory nodes, we assigned the subtype with the highest predicted probability when the predicted uncertainty was below 0.05. We assigned a generic I-UNC (uncertain interneuron) label for all remaining inhibitory nodes. For excitatory nodes we assigned subtypes when the predicted uncertainty was below 0.05. We calculate the overall probability for the pyramidal class by summing the probabilities across all pyramidal cell types. We assigned a thalamocortical label when the predicted thalamocortical probability exceeded the summed pyramidal class probability and the pyramidal subtype with the highest predicted probability otherwise. We assigned a generic P-UNC label for all remaining excitatory nodes.

We assigned cell type classifications to synapses by finding the closest embedding nodes in Euclidean space pre- and post-synaptically and using their cell type labels.

To analyze cell type distributions across the dataset, we randomly sampled 919,500 synapses from the entire dataset. We corrected the native coordinates to orient the dataset vertically between white matter and pia using the standard_transform package (https://github.com/ceesem/standard_transform).

**Axonal sorting**
We skeletonized all of the cells used in the analysis to obtain distances between synapses and the respective soma along the axon. We assigned a distance to each synapse and binned synapses using a width of 20 μm, for which we computed the ratio of synapses predicted as excitatory and inhibitory (Fig. 6j,k) or their inhibitory subtypes (Fig. 6l) while ignoring synapses predicted as uncertain. For each bin we computed the mean and standard error of the mean across all of the cells included in the analysis.

**Reporting summary**
Further information on research design is available in the Nature Portfolio Reporting Summary linked to this article.

## Data availability
All training datasets, precomputed embeddings and image datasets are publicly available. See the online documentation for examples of how to load the individual data products: https://github.com/google-research/connectomics/wiki/SegCLR. The embeddings for the mouse and human datasets are available as sharded csv files: mouse, gs://iarpa_microns/minnie/minnie65/embeddings_m343/segclr_csvzips/README; human, gs://h01-release/data/20220326/c3/embeddings/segclr_csvzips/README. Precomputed annotations are viewable here: https://github.com/google-research/connectomics/wiki/SegCLR#precomputed-embeddings. Pretrained models are available here: https://github.com/google-research/connectomics/wiki/SegCLR#pretrained-embedding-models and https://github.com/google-research/connectomics/wiki/SegCLR#classification-model-training-data-and-pretrained-models. Source data are provided with this paper.

## Code availability
The code is deposited on GitHub under an Apache 2.0 open-source license: https://github.com/google-research/connectomics/tree/main/connectomics/segclr. See the online documentation for more details: https://github.com/google-research/connectomics/wiki/SegCLR#code-release-and-demo-notebooks.

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

## Acknowledgements
The authors thank S. Kornblith, G. Hinton, C. Sun, D. Krishnan and B. Lakshminarayanan for helpful discussions. A.L.B., F.C., C.M.S.-M. and N.M.d.C. were supported by the Intelligence Advanced Research Projects Activity (IARPA) of the Department of Interior/Interior Business

Center (DoI/IBC) through contract number D16PC00004; and by the Allen Institute for Brain Science. The views and conclusions contained herein are those of the authors and should not be interpreted as representing the official policies or endorsements, either expressed or implied, of the funding sources including IARPA, DoI/IBC or the U.S. Government. A.L.B., F.C., C.M.S.-M. and N.m.d.C. thank the Allen Institute founder, P. G. Allen, for his vision, encouragement and support. F.C., C.M.S.-M., A.L.B and N.M.d.C, acknowledge support from NSF NeuroNex 2 award 2014862. J.W.L. acknowledges support from NIH awards U19NS104653, 5UG3MH123386 and U24NS109102.

## Author contributions

S.D., P.H.L., V.J. and M.J. designed SegCLR. S.D., P.H.L. and M.J. implemented SegCLR. J.M.-S., P.H.L. and S.D. created interactive visualizations of SegCLR embeddings. D.R.B. and J.W.L. contributed cell annotations for the h01 dataset. A.L.B., F.C., C.M.S.-M. and N.M.d.C. contributed cell annotations for the MICrONS dataset. P.H.L., M.J., V.J. and S.D. proofread cells in h01. P.H.L., M.J. and S.D. implemented the inference pipeline for large datasets. S.D. and P.H.L. evaluated SegCLR. P.H.L. and S.D. performed the unsupervised analysis. S.D. performed the OOD (out-of-distribution) analysis. S.D. and P.H.L. analyzed the MICrONS data. P.H.L., S.D., V.J. and M.J. wrote the manuscript with contributions from all authors. V.J. led the effort.

## Competing interests

S.D., P.H.L., M.J., J.M.-S. and V.J. are employees of Google LLC, which sells cloud computing services. The other authors declare no competing interests.

## Additional information

**Extended data** are available for this paper at https://doi.org/10.1038/s41592-023-02059-8.

**Correspondence and requests for materials** should be addressed to Viren Jain.

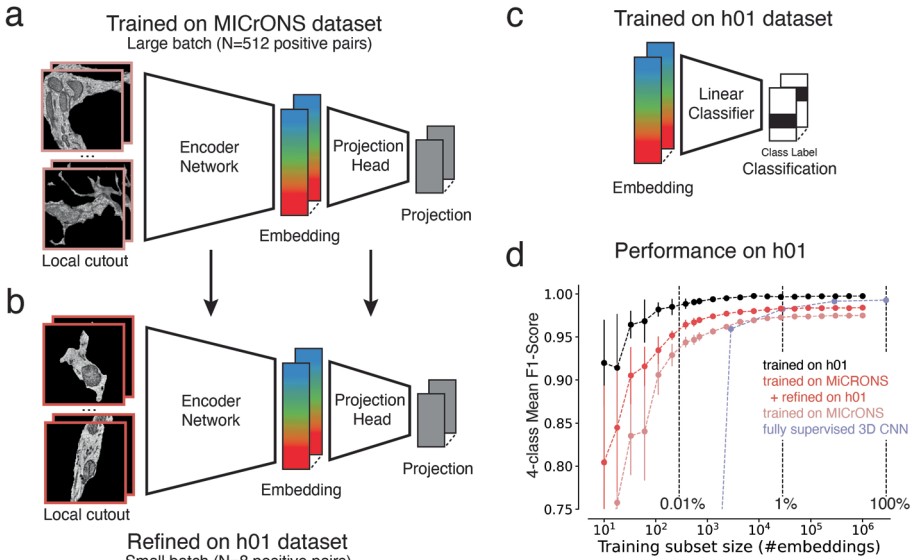

**Extended Data Fig. 1 | Subcompartment classification of SegCLR embeddings trained on a different dataset. a**. The SegCLR model in Fig. 2f was trained on the MICrONS dataset using large training batches. **b**. The model from (a) was further trained (refined) on the h01 dataset with a small batch size on a single machine. **c**. The linear classifier was trained with subcompartment labels for the h01 dataset. **d**. Comparison of linear classifiers trained for the subcompartment

task on the h01 dataset using embeddings from the MICrONS model (gray), the MICrONS model with refinement on the h01 dataset (red), and the h01 model (black). For each training set sample size, mean and standard deviation of multiple subset resamplings is shown (error bars are obscured by the points for larger sample sizes). The light blue line indicates the performance of a fully supervised ResNet-18 classifier trained on the full and subsets of the available training data.

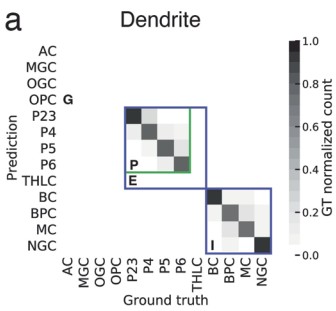

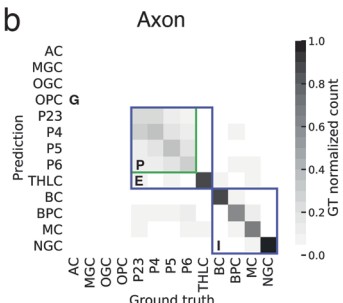

**Extended Data Fig. 2 | Cell type classification of large and small cell fragments via aggregated embeddings. a**. Confusion matrix for the mouse 13-class cell type task at 25 μm aggregation radius. Colored boxes indicate the group of four pyramidal cell types that were collapsed into the 10-class task, and the five excitatory and four inhibitory types collapsed into the 6-class task (similar to Fig. 3h). We restricted this evaluation to dendrites based on the

automated subcompartment classification (Fig. 2) using the same classifier as in Fig. 3 that was trained on the entire ground truth (13-class). Consequently, the test set did not contain any examples of glia subtypes and thalamocortical cells. **b**. As in (a), but restricted to axons. There were no examples of glia subtypes in this test set.

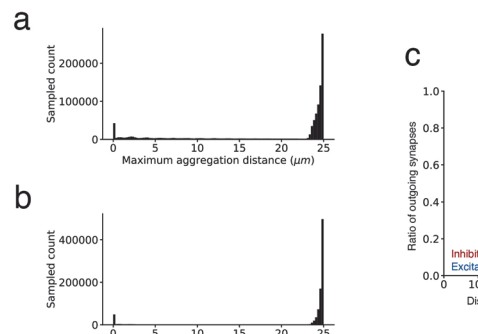

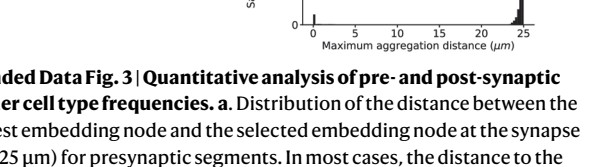

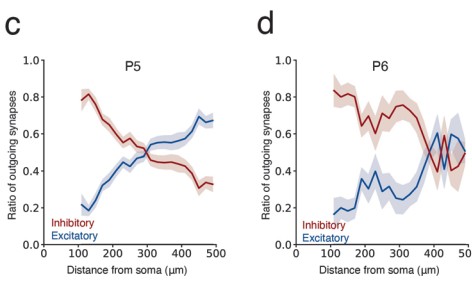

**Extended Data Fig. 3 | Quantitative analysis of pre- and post-synaptic partner cell type frequencies. a**. Distribution of the distance between the farthest embedding node and the selected embedding node at the synapse ($R_{agg}$=25 μm) for presynaptic segments. In most cases, the distance to the farthest embedding node is close to the maximal permitted distance. In a few

cases the presynaptic segment was small, with only a few embedding nodes. **b**. As in (a) for postsynaptic segments. **c.d**. Inhibitory versus excitatory balance of downstream postsynaptic partners with increasing distance along P5 axons (N = 34) (**c**) and P6 axons (N = 6) (**d**).

# Reporting Summary

## Statistics

For all statistical analyses, confirm that the following items are present in the figure legend, table legend, main text, or Methods section.

| n/a | Confirmed | |
|---|---|---|
| ☐ | ☒ | The exact sample size ($n$) for each experimental group/condition, given as a discrete number and unit of measurement |
| ☒ | ☐ | A statement on whether measurements were taken from distinct samples or whether the same sample was measured repeatedly |
| ☒ | ☐ | The statistical test(s) used AND whether they are one- or two-sided *Only common tests should be described solely by name; describe more complex techniques in the Methods section.* |
| ☒ | ☐ | A description of all covariates tested |
| ☒ | ☐ | A description of any assumptions or corrections, such as tests of normality and adjustment for multiple comparisons |
| ☐ | ☒ | A full description of the statistical parameters including central tendency (e.g. means) or other basic estimates (e.g. regression coefficient) AND variation (e.g. standard deviation) or associated estimates of uncertainty (e.g. confidence intervals) |
| ☒ | ☐ | For null hypothesis testing, the test statistic (e.g. $F$, $t$, $r$) with confidence intervals, effect sizes, degrees of freedom and $P$ value noted *Give P values as exact values whenever suitable.* |
| ☒ | ☐ | For Bayesian analysis, information on the choice of priors and Markov chain Monte Carlo settings |
| ☒ | ☐ | For hierarchical and complex designs, identification of the appropriate level for tests and full reporting of outcomes |
| ☒ | ☐ | Estimates of effect sizes (e.g. Cohen's $d$, Pearson's $r$), indicating how they were calculated |

*Our web collection on statistics for biologists contains articles on many of the points above.*

## Software and code

Policy information about availability of computer code

| Data collection | No software was used. |
|---|---|
| Data analysis | Custom computer code in Python 3.10 was developed to train the machine learning method and apply the described computational pipeline on the image volume. The code is open-sourced under a permissive license and available on https://github.com/google-research/connectomics/tree/main/connectomics/segclr |

For manuscripts utilizing custom algorithms or software that are central to the research but not yet described in published literature, software must be made available to editors and reviewers. We strongly encourage code deposition in a community repository (e.g. GitHub). See the Nature Portfolio guidelines for submitting code & software for further information.

## Data

Policy information about availability of data

All manuscripts must include a data availability statement. This statement should provide the following information, where applicable:
- Accession codes, unique identifiers, or web links for publicly available datasets
- A description of any restrictions on data availability
- For clinical datasets or third party data, please ensure that the statement adheres to our policy

All training datasets,p recomputed embeddings,a nd image datasets are publicly available. See the online documentation for examples of how to load the individual data products: https://githu b.com/google-resea rch/con nectom ics/wi ki/SegCLR

March 2021

## Human research participants

Policy information about studies involving human research participants and Sex and Gender in Research.

| | |
|---|---|
| Reporting on sex and gender | not applicable |
| Population characteristics | not applicable |
| Recruitment | not applicable |
| Ethics oversight | not applicable |

Note that full information on the approval of the study protocol must also be provided in the manuscript.

# Field-specific reporting

Please select the one below that is the best fit for your research. If you are not sure, read the appropriate sections before making your selection.

☒ Life sciences ☐ Behavioural & social sciences ☐ Ecological, evolutionary & environmental sciences

For a reference copy of the document with all sections, see nature.com/documents/nr-reporting-summary-flat.pdf

# Life sciences study design

All studies must disclose on these points even when the disclosure is negative.

| | |
|---|---|
| Sample size | Both human and visual cortex datasets are unique large scale acquisitions ("n=1"). We demonstrated our method on many neurons (>100) as outlined in SI1, SI2 |
| Data exclusions | No data was excluded |
| Replication | We replicated our results in one other dataset |
| Randomization | performance tests were carried out multiple times with resampling of training and test sets as described in the methods |
| Blinding | not applicable |

# Reporting for specific materials, systems and methods

We require information from authors about some types of materials, experimental systems and methods used in many studies. Here, indicate whether each material, system or method listed is relevant to your study. If you are not sure if a list item applies to your research, read the appropriate section before selecting a response.

### Materials & experimental systems

| n/a | Involved in the study |
|---|---|
| ☒ | ☐ Antibodies |
| ☒ | ☐ Eukaryotic cell lines |
| ☒ | ☐ Palaeontology and archaeology |
| ☒ | ☐ Animals and other organisms |
| ☒ | ☐ Clinical data |
| ☒ | ☐ Dual use research of concern |

### Methods

| n/a | Involved in the study |
|---|---|
| ☒ | ☐ ChIP-seq |
| ☒ | ☐ Flow cytometry |
| ☒ | ☐ MRI-based neuroimaging |

