## [Peer Review File · Nature Methods]

Peer Review Information

Manuscript Title: Multi-Layered Maps of Neuropil with Segmentation-Guided Contrastive Learning

Corresponding author name(s): Viren Jain

Editorial Notes: n/a

Reviewer Comments & Decisions:

Decision Letter, initial version:

Dear Viren,

Let me first apologize for the unusually long review process. Your Article, "Multi-Layered Maps of Neuropil with Segmentation-Guided Contrastive Learning", has now been seen by three reviewers. As you will see from their comments below, although the reviewers find your work of considerable potential interest, they have raised a number of concerns. We are interested in the possibility of publishing your paper in Nature Methods, but would like to consider your response to these concerns before we reach a final decision on publication.

We therefore invite you to revise your manuscript to address these concerns. Please make sure that the full code and data are made available before resubmission. We also ask that you address the concerns about generalization, as well as the other concerns voiced by the reviewers.

* include a point-by-point response to the reviewers and to any editorial suggestions

* please underline/highlight any additions to the text or areas with other significant changes to facilitate review of the revised manuscript

* address the points listed described below to conform to our open science requirements

* ensure it complies with our general format requirements as set out in our guide to authors at www.nature.com/naturemethods

* resubmit all the necessary files electronically by using the link below to access your home page

[REDACTED]

We hope to receive your revised paper within six weeks. If you cannot send it within this time, please let us know. In this event, we will still be happy to reconsider your paper at a later date so long as nothing similar has been accepted for publication at Nature Methods or published elsewhere.

OPEN SCIENCE REQUIREMENTS

REPORTING SUMMARY AND EDITORIAL POLICY CHECKLISTS

Please note that these forms are dynamic ‘smart pdfs’ and must therefore be downloaded and completed in Adobe Reader. We will then flatten them for ease of use by the reviewers. If you would like to reference the guidance text as you complete the template, please access these flattened versions at <http://www.nature.com/authors/policies/availability.html>.

DATA AVAILABILITY

We strongly encourage you to deposit all new data associated with the paper in a persistent repository where they can be freely and enduringly accessed. We recommend submitting the data to discipline-specific and community-recognized repositories; a list of repositories is provided here:

<http://www.nature.com/sdata/policies/repositories>

All novel DNA and RNA sequencing data, protein sequences, genetic polymorphisms, linked genotype and phenotype data, gene expression data, macromolecular structures, and proteomics data must be deposited in a publicly accessible database, and accession codes and associated hyperlinks must be provided in the “Data Availability” section.

Please include a “Data availability” subsection in the Online Methods. This section should inform readers about the availability of the data used to support the conclusions of your study, including accession codes to public repositories, references to source data that may be published alongside the paper, unique identifiers such as URLs to data repository entries, or data set DOIs, and any other statement about data availability. At a minimum, you should include the following statement: “The data that support the findings of this study are available from the corresponding author upon request”, describing which data is available upon request and mentioning any restrictions on availability. If DOIs are provided, please include these in the Reference list (authors, title, publisher (repository name), identifier, year). For more guidance on how to write this section please see: <http://www.nature.com/authors/policies/data/data-availability-statements-data-citations.pdf>

CODE AVAILABILITY

Please include a “Code Availability” subsection in the Online Methods which details how your custom code is made available. Only in rare cases (where code is not central to the main conclusions of the paper) is the statement “available upon request” allowed (and reasons should be specified).

MATERIALS AVAILABILITY

ORCID

Please do not hesitate to contact me if you have any questions or would like to discuss these revisions further. We look forward to seeing the revised manuscript and thank you for the opportunity to consider your work. I would also like to renew my apologies about the delays.

Best regards,
Nina

Nina Vogt, PhD
Senior Editor
Nature Methods

Reviewers' Comments:

Reviewer #1:

Remarks to the Author:

The manuscript by Dorkenwald et al presents a new method to learn representations of neuron fragments, segmented from electron microscopy volumes. The algorithm starts from a segmentation and implements a procedure similar to the well-established simCLR approach. The two most interesting innovations include the method to sample positive pairs and the addition of the uncertainty prediction for the detection of the out-of-distribution fragments. Similar to contrastive learning as used for self-supervised pre-training on natural images, the proposed approach - termed SegCLR - can learn a rich representation which allows to drastically reduce the complexity of the classifiers and the amount of training data in downstream analysis tasks. As demonstration, the method is applied to supervised cell type prediction tasks and unsupervised exploration of various subclusters in the embedding space.

I find the manuscript both interesting and important. Efficient exploration of enormous EM datasets is an open problem and it's great to see one of the first attempts to solve it (there is an interesting concurrent development for non-neural tissue here: <https://www.biorxiv.org/content/10.1101/2022.05.07.490949v1>, should probably be cited). The unbiased and unsupervised training is also very attractive, I really like the idea of complementing a segmented dataset by such a set of embeddings.

My main concern, however, is that the authors are not giving the community any means to reproduce their findings and build on the top of them. As it stands, their contribution is limited to the embeddings for the 2 big public EM datasets, which will hopefully enable more analysis of these two particular datasets, but nothing else. As the code has not been made available (no, "being prepared for deposition" doesn't count), there is no way to retrain the network on a new dataset. I assume the pretrained one cannot be used for anything else since the authors train two separate networks for their volumes and never mention cross-dataset application. Similarly, from a method developer perspective, it is not clear to me how follow-up research could directly compare with the results presented in the paper. For example, Fig 2E shows that for the mouse dataset the embedding space overlaps for axons and dendrites. This is perfectly fine to demonstrate the proposed approach, but how can one improve on it? Is the human-labeled data available? The authors mention proofreading the segmentations, were the proofread segmentations added back to the public resources? How easy is it to extract the precise training and test datasets used in Fig. 2 from the public resources? On the other hand, arguing from a common, not computationally advanced user's perspective, from a method published in Nature Methods I would normally expect a friendly GUI and not just source code, while here we don't even have source code available. If the aim of the paper is to reach readers beyond the circle of neuroscientists interested in the two public datasets, these points must be addressed.

Other questions:

- is the manually annotated dataset biased towards the cells of the most clear type or is it a dense annotation of a subvolume?
- do you ever use the "classic" contrastive approach of presenting a fragment and its own augmentation as a positive pair?
- how did you decide on the dimensionality of the embedding space? Why not more?
- how dangerous are the segmentation errors? I understand that the learning was performed on a proofread dataset, how much would the embedding change if you apply it on the previous version? I'm asking to get an idea on how useful the method would be with an imperfect segmentation most of us have to live with.
- do you think the difference in performance for subcompartment classification is biological or is it caused by segmentation errors? What do you think it implies for training on the next dataset, e.g. drosophila?

Reviewer #2:

Remarks to the Author:

In this manuscript Dorkenwald et al. propose SegCLR - a method for self-supervised learning of morphological representations of cell fragments in segmented EM data of the neural tissue. Annotating brain volume EM datasets is not only time consuming, but also highly challenging, partially due to the fact that cell processes can extend across vast lengths and often significant cell parts, including somas, could be missing in the imaged volume. To address this problem the authors adapt the latest developments in the field of representation learning for natural images to segmented brain EM data and design an unsupervised pipeline that extracts representations from small cell fragments. The authors further show how such representations can be used for multiple annotation tasks.

I find the proposed method to be technically sound and promising for analyzing electron microscopy imaging data. The fact that classifying cellular subcompartments requires 4000 times less training data when using SegCLR representations shows the potential of the method to significantly speed up the analysis of the brain EM volumes. Furthermore, reaching high accuracy in classifying cell types even on short length cell fragments that are challenging for human annotators would allow for more unbiased larger scale analysis of brain circuits. However, I found the unsupervised data exploration part much less convincing, since most of it heavily relies on extensive manual annotations. Moreover, most of the analysis presented has been performed only on the manually proofread parts of the data, raising a question of how well the method would generalize on the whole dataset. Furthermore, the clarity of the manuscript could be improved. Considering this, I believe the following points should be addressed before the paper is considered for publishing:

Major:

- The flow of the introduction could be improved. For example, it would be easier to follow if self-supervised/contrastive learning was explained before introducing SegCLR. Moreover, it would be helpful if the contributions of the method were listed in one or sequential paragraphs. I guess that moving the paragraph 3 after the paragraph introducing self-supervised learning would already greatly facilitate understanding what the method does. Comparison to the previous work should rather focus on the novelty of the presented method and the new types of analysis it enables, as opposed to listing the types of analysis not done in the previous work. For example, consider the sentence "Schubert et al. trained cell representations using a triplet loss, but it was not reported whether these representations are suitable for downstream analyses". Schubert et al. did show the usefulness of their representations on multiple tasks. However, using 3D EM data clearly results in richer representations than using 2D projections or shape only, and I feel like the authors should rather focus on this strength. Finally, the concurrent work on self-supervised/contrastive learning on EM data (Wilson et al., 2022, Zinchenko et al. 2023) should also be cited for completeness.

- I found the figures to be a bit misleading, because except for Figure 1b only the cell skeletons are shown, not the underlying EM data. Especially since the authors mostly showed a much bigger cutout than was used for extracting representations, it often gave a false impression of the task being rather trivial to solve. However, it is impressive that the cell types and cellular subcompartments can be predicted from small fragments, so showing the actual fragments could not only help get a feeling of how different the underlying texture is, but also convey a much stronger message.
- “For evaluation of local cell type classification (Fig. 4), it was important to have cells with minimal reconstruction merge errors in the ground truth labeled set.” I would argue that such a requirement greatly diminishes the value of the presented results. Since manual segmentation proofreading and correction is mostly more time-consuming than cell annotation, I would expect the method to be often applied to datasets where segmentation errors are not rare. Thus, it would be extremely beneficial if the authors could adjust their cell classification pipeline to deal with merge errors as well. For example, given the promising results in the Figure 5g, the authors could use their out-of-distribution input detection to locate merge errors that would further increase the value of the method.
- Figure 2: why is “the performance of a fully supervised ResNet-18 classifier trained on the full available training data” only shown for the human data, but not for the mouse data? The notably worse performance of the method on the mouse data in comparison to the human data should also be explained.
- “We used unsupervised UMAP projection to visualize samples of embeddings in the human and mouse datasets, and readily observed separate regions in UMAP space for glia versus neurons, and for axons, dendrites, and somas”. The fact that there were separate regions in the UMAP space should not be used to claim that SegCLR is useful for unsupervised exploration, because these UMAP plots in the first place were generated from a representative subsample of embeddings, where the definition of “representative” is based on manual annotations. If it is computationally infeasible to use UMAP on all of the generated embeddings, a random subsample of the embeddings should be selected to run UMAP on.
- Similarly, a specific set of cells that was proofread and labeled by human experts should not be used to showcase using SegCLR embeddings for unsupervised data exploration. If the goal is to focus on a smaller subset of cells, predictions of the cell type classifier could be used to extract a specific subset.
- Furthermore, the semi-automatic pipeline of cluster extraction for unsupervised exploration of mouse visual cortex layer-5 pyramidal cells requires too many manual choices to be considered unsupervised: 3 clusters were defined based on the UMAP components 2 and 3 by running k-means with 25 groups on the first 5 UMAP components and assigning them to the visual UMAP clusters. I feel like this complexity could be reduced by using better clustering methods in the first place. While k-means is fast and efficient, it is not the best option for clustering sparse high-dimensional data. I would expect better results could be obtained using density-based HDBSCAN on several UMAP components or graph-based clustering methods directly on the representations, for example, the Louvain or the Leiden algorithms commonly used in the single-cell transcriptomics field.
- Code availability: this section should include the link to the used code.

Minor:

- The gray color scale for heatmaps is a bit difficult to read.
- Figure 3a: what is WM? It would also be great if the scale bars of the upper and the lower row were vertically aligned.
- Figure 6: would be helpful if the abbreviations were explained in the figure legends.
- Methods: "For comparison, we also trained a fully-supervised subcompartment classifier directly on voxel inputs using an identical 3d ResNet-18 architecture and input configuration (photometric augmentation was omitted and random 3d rotations were added)". What was the reason to omit photometric augmentation? It seems important given the dataset with varying intensities.
- Methods: "For each fold we found the uncertainty threshold that maximized the F1-Score of the in-distribution vs out-of-distribution task." I feel like using one threshold for all folds would be more appropriate. Or at least these thresholds should be reported to give an estimate of their variance.
- Methods: "Next, we labeled nodes as uncertain where the predicted uncertainty was above 0.45." How was this threshold determined? In case of no evaluation data available, would it not be more intuitive to use 0.5?
- Methods: "Here, we assigned a thalamocortical label when the predicted thalamocortical probability exceeded the summed probability across all pyramidal cell types and the pyramidal subtype with the highest predicted probability otherwise". This has to be justified.
- It would be helpful if the Methods section contained direct links to access the data when it's mentioned for the first time, including links to the used datasets, generated representations and cell type annotations, as well as the used code.

Due to my limited proficiency in neurobiology, the biological conclusions of the presented analyses are outside the scope of my expertise.

References

- Wilson, A. M., & Babadi, M. (2022). Uncovering features of synapses in primary visual cortex through contrastive representation learning. *bioRxiv*, 2022-06.
- Zinchenko, V., Hugger, J., Uhlmann, V., Arendt, D., & Kreshuk, A. (2023). MorphoFeatures for unsupervised exploration of cell types, tissues, and organs in volume electron microscopy. *Elife*, 12, e80918.

Reviewer #3:

Remarks to the Author:

The manuscript by Dorckenwald and Li et al. describes an extension to the unsupervised learning method SimCLR and its application to two large electron microscopy (EM) connectomics datasets. The extension

consists in the consideration of additional positive sample pairs from nearby locations within the same segment, and thus requires a prior segmentation of the EM volume. The embeddings learnt with this method (SegCLR) are shown to be very information rich: the embeddings allow classification of cellular subcompartments and cell types when used as input to a shallow classification network, requiring less annotated samples than fully supervised methods on the same image data.

The extension to SimCLR is rather simple: in SimCLR, positive pairs are obtained through augmentations (e.g., rotations, elastic deformations, intensity changes). In SegCLR, additional positive pairs are sampled within a threshold distance within the same object, as given by a segmentation. The significance of this work lies mostly in the application of the extension to two connectomics datasets and the demonstration that the learnt embeddings aid substantially in solving downstream analysis tasks (e.g., cell typing of downstream partners from local post-synaptic image patches). The presented experiments convincingly support the claims that SegCLR embeddings allow cellular subcompartment classification and cell typing on two large EM datasets.

1. The claim that SegCLR reduces the amount of labelled data required by a factor of 4,000 stems from the 4-class subcompartment task on the human dataset: "On the 4-class subcompartment task, the embedding-based classification matches the performance of direct supervised training while requiring roughly 4,000 times less labeled training data (10-run median F1-Score, ~400 examples total)". This claim is not sufficiently backed up.

The fully supervised baseline has only been trained on the full dataset. Judging from the progression of F1-scores on the embedding-based classifications, not all of the provided training data might be needed. This is not accounted for.

The fully supervised baseline should also be evaluated on randomly sampled subset of different sizes (as done for SegCLR) to see at what point the baseline performance plateaus. Statements about the amount of training data needed for either method should be made with respect to the obtained F1-score (e.g., "SegCLR requires X-fold less training data than a fully supervised baseline to reach an F1-score of Y").

2. The input to the SegCLR network is a downsampled volume (32-40nm resolution). I would appreciate an elaboration on the rationale behind the downsampling, especially since the discussion itself mentions that the lower resolution impedes SegCLR's capabilities to analyze finer ultrastructure like vesicles (which is presumably relevant for other downstream analysis tasks beyond compartment and cell type classification). Did the authors apply SegCLR at the native resolution but found the results to be inferior for the presented analysis tasks? This would be worth reporting. Or are there technical considerations that would favor a near isotropic input (e.g., rotation augmentations in $SO(3)$) that are crucial for learning embeddings?

3. Figure 2f does not contain a fully supervised baseline but is otherwise similar to Figure 2d. My understanding is that a fully supervised method can be trained on the same locations for which SegCLR produced embeddings. Please provide the baseline comparison here as well, in particular since it would provide evidence for the claim that SegCLR requires less training data.

Minor points:

* The y-axes in Fig. 2d should be rescaled to better see differences.

* Discussion of reference [32] (Huang et al., 2020) could state that SimCLR was used there as well.

* A more recent (and presumably more relevant for EM ultrastructure) reference for attribution methods instead of [57] (Syndararajan et al., 2017) is the attribution method by Eckstein et al.:

Eckstein, Nils, et al. "Discriminative Attribution from Paired Images." Computer Vision—ECCV 2022 Workshops: Tel Aviv, Israel, October 23–27, 2022, Proceedings, Part IV. Cham: Springer Nature Switzerland, 2023.

* Methods, below Equation 1: "512 pairs" -> "512 positive pairs"

* Methods, "Unsupervised exploration of mouse [...]": References to Fig. 2 should be to Fig. 4

* Methods, "Automated analysis of synaptic partners": "we λ " -> "we set λ "

Author Rebuttal to Initial comments

We thank the reviewers for their detailed and constructive feedback!

We made several changes to the manuscript and added experiments to address the questions brought forward in the review.

Overview of the main changes to the submission:

1. In order to address the question of whether readers can benefit from the SegCLR method for their own datasets in the absence of performing the intensive training procedure to learn new embeddings, we performed several new experiments. First, we applied the MICrONS model to the h01 data and retrained the (cheap) compartment classification model and found that the embeddings produced by a SegCLR model trained on a different dataset still performed well. Second, we describe how to fine-tune the pretrained models using only a single accelerator and show that this further improves performance (Fig. 2, Sup. Fig. 2).
2. To further tighten the critical issue of comparison to fully supervised methods, we trained additional fully-supervised compartment models on subsets of the training data to compare their scaling behavior with that of classifiers trained on SegCLR embeddings (Fig. 2). In line with our previous findings, we find that SegCLR embeddings produce remarkably high performing classifiers (>0.95 F1-Score) even in exceptionally low regimes of labeled examples (hundreds) while maintaining complete parity with fully supervised methods at 400-fold fewer labels.

While running these new experiments, we discovered that a small fraction of h01 skeletons (41 out of 853) contributed a small number of examples (1026 out of 672297 total test set examples) to both the training and test sets. For simplicity we therefore dropped all 41 contaminated skeletons (11% of the total test examples) from the test set in the new analysis.

3. We provide open-source code for SegCLR under a permissive license, along with pretrained models and notebooks demonstrating how to train, load, evaluate, and apply SegCLR models. We created documentation that describes the code and data here: <https://github.com/google-research/connectomics/wiki/SegCLR>
4. We reorganized the introduction for clarity and adjusted the text to resolve points that were flagged as unclear during the review.

Below, we respond to specific reviewer remarks in *italics*.

Reviewer #1:

Remarks to the Author:

The manuscript by Dorkenwald et al presents a new method to learn representations of neuron fragments, segmented from electron microscopy volumes. The algorithm starts from a segmentation and implements a procedure similar to the well-established simCLR approach. The two most interesting innovations include the method to sample positive pairs and the addition of the uncertainty prediction for the detection of the out-of-distribution fragments. Similar to contrastive learning as used for self-supervised pre-training on natural images, the proposed approach - termed SegCLR - can learn a rich representation which allows to drastically reduce the complexity of the classifiers and the amount of training data in downstream analysis tasks. As demonstration, the method is applied to supervised cell type prediction tasks and unsupervised exploration of various subclusters in the embedding space.

I find the manuscript both interesting and important. Efficient exploration of enormous EM datasets is an open problem and it's great to see one of the first attempts to solve it (there is an interesting concurrent development for non-neural tissue here: <https://www.biorxiv.org/content/10.1101/2022.05.07.490949v1>, should probably be cited). The unbiased and unsupervised training is also very attractive, I really like the idea of complementing a segmented dataset by such a set of embeddings.

My main concern, however, is that the authors are not giving the community any means to reproduce their findings and build on the top of them. As it stands, their contribution is limited to the embeddings for the 2 big public EM datasets, which will hopefully enable more analysis of these two particular datasets, but nothing else. As the code has not been made available (no, "being prepared for deposition" doesn't count), there is no way to retrain the network on a new dataset. I assume the pretrained one cannot be used for anything else since the authors train two separate networks for their volumes and never mention cross-dataset application. Similarly, from a method developer perspective, it is not clear to me how follow-up research could directly compare with the results presented in the paper. For example, Fig 2E shows that for the mouse dataset the embedding space overlaps for axons and dendrites. This is perfectly fine to demonstrate the proposed approach, but how can one improve on it? Is the human-labeled data available? The authors mention proofreading the segmentations, were the proofread segmentations added back to the public resources? How easy is it to extract the precise training and test datasets used in Fig. 2 from the public resources? On the other hand, arguing from a common, not computationally advanced user's perspective, from a method published in Nature Methods I would normally expect a friendly GUI and not just source code, while here we don't even have source code available. If the aim of the paper is to reach readers beyond the circle of neuroscientists interested in the two public datasets, these points must be addressed.

Generalizability: While we test our approach on two large EM datasets, for which we make the resulting embeddings and pretrained models available, there are no limitations in datasets the SegCLR method can be applied to, as long as an instance segmentation is available.

New experiment: Thank you for the suggestion about cross-dataset applications. To test the generalizability/transferability of a pretrained model, we cross-predicted embeddings on the h01 dataset using the model trained on the MICrONS dataset, and then repeated the subcompartment classification task. We found that a linear classifier trained on these embeddings performed well, although slightly below the fully supervised model and the linear classifier trained on embeddings predicted from a model trained on the h01 dataset (Sup. Fig. 2).

We further tested whether minimal model weight refinement (continuing training on a smaller scale with a single accelerator) on the h01 dataset could improve the MICrONS pretrained model embeddings. We found that the refinement was able to close the performance gap between the MICrONS-derived and h01-derived embeddings by 40% on the compartment task. This shows the utility of pretrained models for other datasets, beyond the generalizability of the method itself.

Code and Replication: We now make all code and data publicly available. Readers can get the same training and test sets and use our subsampling code to recreate the same training and testing scenarios. This enables readers to train SegCLR models on their own datasets.

We also made an interactive browser to explore the h01 embeddings available here: <https://h01-release.storage.googleapis.com/embeddings.html>

Other questions:

- is the manually annotated dataset biased towards the cells of the most clear type or is it a dense annotation of a subvolume?

MICrONS: The annotations were derived by combining data from existing cell typing efforts with information about their proofread status. Cell typing was done in a largely unbiased way; all cells within a cylinder were labeled (MICrONS consortium, 2021; Schneider-Mizell et al., 2023) and some cells of interest in other places were added as well. All inhibitory cells were proofread by the MICrONS effort, excitatory cells were only partially proofread and there is a small bias towards Layer 2/3 pyramidal cells because they were of larger interest but not because they were easier to classify. The dataset includes cells of all major cell types except for chandelier cells. There happened to be no chandelier cell in the spatial column.

h01: Most cells with cell body in the dataset were cell typed as excitatory, inhibitory, or glia (Shapson-Coe et al., 2021) and as such were unbiased. The h01 effort also added labels for glia subtypes where the reconstruction permitted. The classification of microglia and OPCs was difficult for experts without having a more complete reconstruction. We selected candidates for proofreading from the neuron classes based on the completeness of the automated reconstruction and performed proofreading of them ourselves.

The cell type counts in our ground truth are necessarily not representative of the brain as a whole. We accounted for this during evaluation by sampling uniformly across cell types.

- do you ever use the "classic" contrastive approach of presenting a fragment and its own augmentation as a positive pair?

In the presented work, we do not use the classic approach. The SegCLR project started with using the classic approach, i.e. we applied various augmentations to a cutout to create positive pairs, but we found that the resulting embeddings performed poorly. In some cases we even found that the augmentations were harmful (e.g. crop and resize, the best performing one in the SimCLR paper). We believe this is because these augmentations change important features about the morphology to which downstream tasks are not invariant. We subsequently added the segmentation guidance and reduced image distortions to a minimum (flips, intensity changes).

- how did you decide on the dimensionality of the embedding space? Why not more?

In initial experiments, we observed a slight increase in performance when increasing the embedding dimensionality from 16, to 32, to 64 dimensions. More dimensions require proportionally more storage and make downstream analyses slower. We settled on 64 dimensions as a good tradeoff to keep overall data volumes manageable. We added a comment about this to the methods.

- how dangerous are the segmentation errors? I understand that the learning was performed on a proofread dataset, how much would the embedding change if you apply it on the previous version? I'm asking to get an idea on how useful the method would be with an imperfect segmentation most of us have to live with.

Our networks were trained on largely unproofread data. For h01, no proofreading was done on the data used for training the embedding models. For MICrONS, the dataset contained proofreading but we did not specifically select for these cells during training. Given that there are millions of segments in the dataset, the proofreading of the ~1,000 cells likely did not impact training much. Still, we agree it is an interesting question to see whether training SegCLR models on proofread data would improve performance or if the decrease in overall training data would reduce performance more.

- do you think the difference in performance for subcompartment classification is biological or is it caused by segmentation errors? What do you think it implies for training on the next dataset, e.g. drosophila?

The compartment classification task on the h01 data included astrocytic processes (mirroring Li et al., 2020) whereas the evaluation on the MICrONS data only included axon, dendrite, and soma. The astrocyte class tends to be among the higher performing

ones which inflates the absolute performance numbers making a direct comparison of the F1-Scores difficult. Beyond the difference in the evaluation task, there are several differences between the datasets that could explain the performance gap. The automated segmentation for h01 generally contained many fewer merge errors which might be beneficial for SegCLR training as fewer wrong positive pairs are generated for training. Sample preparation, species, and ground truth quality could play a role as well. We do not see a reason why our approach would not generalize to other datasets such as those from Drosophila.

Reviewer #2:

Remarks to the Author:

In this manuscript Dorkenwald et al. propose SegCLR - a method for self-supervised learning of morphological representations of cell fragments in segmented EM data of the neural tissue. Annotating brain volume EM datasets is not only time consuming, but also highly challenging, partially due to the fact that cell processes can extend across vast lengths and often significant cell parts, including somas, could be missing in the imaged volume. To address this problem the authors adapt the latest developments in the field of representation learning for natural images to segmented brain EM data and design an unsupervised pipeline that extracts representations from small cell fragments. The authors further show how such representations can be used for multiple annotation tasks.

I find the proposed method to be technically sound and promising for analyzing electron microscopy imaging data. The fact that classifying cellular subcompartments requires 4000 times less training data when using SegCLR representations shows the potential of the method to significantly speed up the analysis of the brain EM volumes. Furthermore, reaching high accuracy in classifying cell types even on short length cell fragments that are challenging for human annotators would allow for more unbiased larger scale analysis of brain circuits. However, I found the unsupervised data exploration part much less convincing, since most of it heavily relies on extensive manual annotations. Moreover, most of the analysis presented has been performed only on the manually proofread parts of the data, raising a question of how well the method would generalize on the whole dataset. Furthermore, the clarity of the manuscript could be improved. Considering this, I believe the following points should be addressed before the paper is considered for publishing:

We applied SegCLR to the entire datasets presented in the manuscript and made that data publicly available. We demonstrate the generalizability of the SegCLR embeddings by evaluating the classification of small cutouts, including large-scale classification of synaptic partners (fig. 6). We also added out-of-distribution detection (fig. 5) to reduce erroneous classifications on examples belonging to unseen classes. Our emphasis on manually proofread data in some sections is in order to have fully reviewed ground truth against which to evaluate the performance of our method, but this is not a major requirement for most applications; see below.

Major:

- The flow of the introduction could be improved. For example, it would be easier to follow if self-supervised/contrastive learning was explained before introducing SegCLR. Moreover, it would be helpful if the contributions of the method were listed in one or sequential paragraphs. I guess that moving the paragraph 3 after the paragraph introducing self-supervised learning would already greatly facilitate understanding what the method does. Comparison to the previous work should rather focus on the novelty of the presented method and the new types of analysis it enables, as opposed to listing the types of analysis not done in the previous work. For example, consider the sentence "Schubert et al. trained cell representations using a triplet

loss, but it was not reported whether these representations are suitable for downstream analyses". Schubert et al. did show the usefulness of their representations on multiple tasks. However, using 3D EM data clearly results in richer representations than using 2D projections or shape only, and I feel like the authors should rather focus on this strength. Finally, the concurrent work on self-supervised/contrastive learning on EM data (Wilson et al., 2022, Zinchenko et al. 2023) should also be cited for completeness.

We improved the introduction largely following the suggestions outlined and added the mentioned citations.

- I found the figures to be a bit misleading, because except for Figure 1b only the cell skeletons are shown, not the underlying EM data. Especially since the authors mostly showed a much bigger cutout than was used for extracting representations, it often gave a false impression of the task being rather trivial to solve. However, it is impressive that the cell types and cellular subcompartments can be predicted from small fragments, so showing the actual fragments could not only help get a feeling of how different the underlying texture is, but also convey a much stronger message.

While we use skeletons to extract positive training pairs, the model is not presented with the skeleton itself. In figures 1 and 2 we do not show skeletons, but we do show the segment mesh more often than the EM input data, which is hard to visualize in 3d. We show the field of view of the network as indicated by the blue box on the mesh in 1a, the EM cutout in 1b, the dark shaded areas on the meshes in 1c,d and the boxes on top of the meshes in 2b. We use the skeletons to aggregate the embeddings through averaging and show examples of the context sizes in 3b. We added additional notes in the text to emphasize that EM input data is not always shown alongside the segment masks.

- "For evaluation of local cell type classification (Fig. 4), it was important to have cells with minimal reconstruction merge errors in the ground truth labeled set." I would argue that such a requirement greatly diminishes the value of the presented results. Since manual segmentation proofreading and correction is mostly more time-consuming than cell annotation, I would expect the method to be often applied to datasets where segmentation errors are not rare. Thus, it would be extremely beneficial if the authors could adjust their cell classification pipeline to deal with merge errors as well. For example, given the promising results in the Figure 5g, the authors could use their out-of-distribution input detection to locate merge errors that would further increase the value of the method.

(The reference to Figure 4 instead of Figure 3 in our previous revision was an error.) We agree that SegCLR will most commonly be used on datasets containing substantial segmentation errors, and we demonstrate that it is applicable to these settings. Importantly, the cited quote from the methods refers to the importance of minimal merge errors only in order to have ground truth to evaluate classification performance against for purposes of the paper. To be able to demonstrate cell typing of local fragments, we

required largely complete and proofread cells for which an expert could assign a cell type label with high confidence. Then, by cutting these cells into fragments we tested the prediction of local fragments and evaluated how large of a context is required. By showing that small contexts are sufficient, we showed that SegCLR can be applied to incomplete and erroneous reconstructions which we demonstrate in Figure 6. We added notes in the text to clarify this point.

We also agree that local SegCLR predictions would be applicable to tasks such as merge error detection. We explore this type of application in related prior work (Li et al. 2020) and note it as an area for future research in our discussion here.

- Figure 2: why is “the performance of a fully supervised ResNet-18 classifier trained on the full available training data” only shown for the human data, but not for the mouse data? The notably worse performance of the method on the mouse data in comparison to the human data should also be explained.

The compartment classification task on the h01 data included astrocytic processes (mirroring Li et al., 2020) whereas the evaluation of the MICrONS data only included axon, dendrite, soma. The astrocyte class tends to be among the higher performing ones which increases the absolute performance number (mean across all classes) making a direct comparison of the F1-Scores difficult. Beyond the difference in the task, there are several differences between the datasets that could explain the performance gap. The automated segmentation for h01 generally contained many fewer merge errors which might be beneficial for SegCLR training as fewer wrong positive pairs are generated for training. Sample preparation, species, and ground truth quality could play a role as well. Since training fully supervised models takes considerable time and resources, we focused efforts on the h01 dataset (including extended supervised training for the latest revision). We do not have reason to expect the results on MICrONS to differ substantially.

- “We used unsupervised UMAP projection to visualize samples of embeddings in the human and mouse datasets, and readily observed separate regions in UMAP space for glia versus neurons, and for axons, dendrites, and somas”. The fact that there were separate regions in the UMAP space should not be used to claim that SegCLR is useful for unsupervised exploration, because these UMAP plots in the first place were generated from a representative subsample of embeddings, where the definition of “representative” is based on manual annotations. If it is computationally infeasible to use UMAP on all of the generated embeddings, a random subsample of the embeddings should be selected to run UMAP on.

- Similarly, a specific set of cells that was proofread and labeled by human experts should not be used to showcase using SegCLR embeddings for unsupervised data exploration. If the goal is to focus on a smaller subset of cells, predictions of the cell type classifier could be used to extract a specific subset.

- Furthermore, the semi-automatic pipeline of cluster extraction for unsupervised exploration of mouse visual cortex layer-5 pyramidal cells requires too many manual choices to be considered unsupervised: 3 clusters were defined based on the UMAP components 2 and 3 by running

k-means with 25 groups on the first 5 UMAP components and assigning them to the visual UMAP clusters. I feel like this complexity could be reduced by using better clustering methods in the first place. While k-means is fast and efficient, it is not the best option for clustering sparse high-dimensional data. I would expect better results could be obtained using density-based HDBSCAN on several UMAP components or graph-based clustering methods directly on the representations, for example, the Louvain or the Leiden algorithms commonly used in the single-cell transcriptomics field.

In Figure 1, we randomly sampled points from the entire dataset and showed that the embeddings cluster in regions related to axons, dendrites, astrocytes, and soma. The same can be observed in Figure 4 where we only considered a subset of the cells. For the unsupervised exploration shown in Figure 4, we wanted to demonstrate how SegCLR's embedding can be used to guide cell type discovery. We believe that such a process, currently, requires an expert to judge which signals in the data (here: embeddings) are related to "cell types." The subtyping of Layer 5 pyramidal cells was chosen because it resembled a real research question that came up in the analysis of this data within the MICrONS consortium.

Similar to the evaluation in Figure 3, we required somewhat complete and proofread neurons to demonstrate the correctness of our resulting unsupervised clustering. Without the ability to relate the clusters to proofread neurons, we could not show that the discovered clusters are corroborated by other features in the neuronal morphology.

We believe that much more work can and should be done on applying unsupervised methods to embeddings created by SegCLR and other methods. Within the scope of this paper and the space available to explore unsupervised methods on the embeddings, we chose one of the best known and widely available methods (k-means) to demonstrate the applicability of SegCLR's embeddings. We agree that the mentioned clustering and dimensionality reduction algorithms are good candidates to explore on these embeddings.

- Code availability: this section should include the link to the used code.

Fixed

Minor:

- The gray color scale for heatmaps is a bit difficult to read.

We explored different colormaps for the confusion matrices, but given that the quantity being represented is 1-dimensional we feel a gray scale is reasonable and avoids any issue of illegibility for colorblind readers (e.g. if multiple colors were introduced)..

- Figure 3a: what is WM? It would also be great if the scale bars of the upper and the lower row were vertically aligned.

Fixed; thank you!

- Figure 6: would be helpful if the abbreviations were explained in the figure legends.

Added.

- Methods: "For comparison, we also trained a fully-supervised subcompartment classifier directly on voxel inputs using an identical 3d ResNet-18 architecture and input configuration (photometric augmentation was omitted and random 3d rotations were added)". What was the reason to omit photometric augmentation? It seems important given the dataset with varying intensities.

To test whether photometric augmentation could improve the fully-supervised subcompartment classifier, we retrained the model while using SegCLRs photometric augmentation (and removing the extra 3d rotations). We found that the resulting network performed somewhat worse than the one included in the paper and did not include it in the submission. The task of the supervised classifier is different from the task of the embedding model, so it is perhaps not surprising that the relative value of different augmentations differs.

- Methods: "For each fold we found the uncertainty threshold that maximized the F1-Score of the in-distribution vs out-of-distribution task." I feel like using one threshold for all folds would be more appropriate. Or at least these thresholds should be reported to give an estimate of their variance.

The uncertainty is not normalized and thresholds are not strictly comparable and interpretable because every fold contains a different training set (albeit sampled from the same underlying data) and the uncertainty measurement is influenced by the spread of the data and the confidence of the classifier for the task. Additionally, the thresholds are not bound to a specific upper limit.

- Methods: "Next, we labeled nodes as uncertain where the predicted uncertainty was above 0.45." How was this threshold determined? In case of no evaluation data available, would it not be more intuitive to use 0.5?

The uncertainty is not normalized, so a 0.5 value doesn't hold inherent significance. In this case uncertainty ranged up to 0.6. The uncertainty threshold can be set at different points depending on the needs of the analysis (more labels, less conservative or fewer labels, more conservative). In our case, we sampled ~100 synapses and interrogated their correctness. "Out of distribution" more often than not also meant cases with artifacts (merged axons, tiny axon fragments, EM image artifacts, ...). We labeled such cases and used that to set a threshold at 0.45. We now mention that in the methods.

- Methods: "Here, we assigned a thalamocortical label when the predicted thalamocortical probability exceeded the summed probability across all pyramidal cell types and the pyramidal subtype with the highest predicted probability otherwise". This has to be justified.

The summed probability of all pyramidal cell classes represents the probability for the pyramidal class as a whole (see hierarchy in Fig 3). We clarified this in the methods.

- It would be helpful if the Methods section contained direct links to access the data when it's mentioned for the first time, including links to the used datasets, generated representations and cell type annotations, as well as the used code.

We added a methods link to an online documentation page summarizing all data products and code in one place with example notebooks for how to load and use them:
<https://github.com/google-research/connectomics/wiki/SegCLR>

Due to my limited proficiency in neurobiology, the biological conclusions of the presented analyses are outside the scope of my expertise.

References

- Wilson, A. M., & Babadi, M. (2022). Uncovering features of synapses in primary visual cortex through contrastive representation learning. bioRxiv, 2022-06.
- Zinchenko, V., Hugger, J., Uhlmann, V., Arendt, D., & Kreshuk, A. (2023). MorphoFeatures for unsupervised exploration of cell types, tissues, and organs in volume electron microscopy. Elife, 12, e80918.

Reviewer #3:

Remarks to the Author:

The manuscript by Dorkenwald and Li et al. describes an extension to the unsupervised learning method SimCLR and its application to two large electron microscopy (EM) connectomics datasets. The extension consists in the consideration of additional positive sample pairs from nearby locations within the same segment, and thus requires a prior segmentation of the EM volume. The embeddings learnt with this method (SegCLR) are shown to be very information rich: the embeddings allow classification of cellular subcompartments and cell types when used as input to a shallow classification network, requiring less annotated samples than fully supervised methods on the same image data.

The extension to SimCLR is rather simple: in SimCLR, positive pairs are obtained through augmentations (e.g., rotations, elastic deformations, intensity changes). In SegCLR, additional positive pairs are sampled within a threshold distance within the same object, as given by a segmentation. The significance of this work lies mostly in the application of the extension to two connectomics datasets and the demonstration that the learnt embeddings aid substantially in solving downstream analysis tasks (e.g., cell typing of downstream partners from local post-synaptic image patches). The presented experiments convincingly support the claims that SegCLR embeddings allow cellular subcompartment classification and cell typing on two large EM datasets.

1. The claim that SegCLR reduces the amount of labelled data required by a factor of 4,000 stems from the 4-class subcompartment task on the human dataset: "On the 4-class subcompartment task, the embedding-based classification matches the performance of direct supervised training while requiring roughly 4,000 times less labeled training data (10-run median F1-Score, ~400 examples total)". This claim is not sufficiently backed up.

The fully supervised baseline has only been trained on the full dataset. Judging from the progression of F1-scores on the embedding-based classifications, not all of the provided training data might be needed. This is not accounted for.

The fully supervised baseline should also be evaluated on randomly sampled subset of different sizes (as done for SegCLR) to see at what point the baseline performance plateaus. Statements about the amount of training data needed for either method should be made with respect to the obtained F1-score (e.g., "SegCLR requires X-fold less training data than a fully supervised baseline to reach an F1-score of Y").

Thank you for this suggestion. We added new experiments to test the scaling behavior of the fully supervised model with the amount of training labels. We found that the fully supervised model performs very poorly for small training sets (<.1%, N<2846), likely due to overfitting, whereas models trained on the embeddings still performed well with few samples. We now use the 10% training data point as reference for calculating ground truth reduction. The fully supervised model plateaus at that point with little improvement

in the 100% case. We adjusted the text to include references to performance numbers for which the reduction of ground truth amount was described.

2. The input to the SegCLR network is a downsampled volume (32-40nm resolution). I would appreciate an elaboration on the rationale behind the downsampling, especially since the discussion itself mentions that the lower resolution impedes SegCLR's capabilities to analyze finer ultrastructure like vesicles (which is presumably relevant for other downstream analysis tasks beyond compartment and cell type classification). Did the authors apply SegCLR at the native resolution but found the results to be inferior for the presented analysis tasks? This would be worth reporting. Or are there technical considerations that would favor a near isotropic input (e.g., rotation augmentations in SO(3)) that are crucial for learning embeddings?

We used the 32/40 nm resolution after referring to the experiments in Li et al., 2020 in which this approximate resolution was found to be effective for similar tasks. Using higher resolutions reduces the field of view of the network (in nanometers) that can be fit onto the same accelerator memory. Therefore, the addition of higher resolutions represents a tradeoff. Embeddings generated with higher resolution data will likely perform better on tasks that rely more on fine details in the EM data whereas embeddings generated with lower resolutions likely perform better on tasks that leverage the shape of the cell over larger distances. We think that multi-resolution approaches will combine these advantages provided by each resolution. This could be achieved simply e.g. by training multiple models and concatenating the embeddings. We found preliminarily that this could improve performance on the tasks presented in the paper, but did not have time or space to explore it comprehensively and decided not to include it for simplicity given the strong performance of the single resolution tested here. Future work should evaluate this component of SegCLR.

3. Figure 2f does not contain a fully supervised baseline but is otherwise similar to Figure 2d. My understanding is that a fully supervised method can be trained on the same locations for which SegCLR produced embeddings. Please provide the baseline comparison here as well, in particular since it would provide evidence for the claim that SegCLR requires less training data.

We invested into an extension of our experiments on the h01 dataset by adding fully supervised model evaluations at reduced ground truth to strengthen the point of superior performance, especially in the low sample regime. Training these fully supervised models requires ~2 weeks of training per model with potential redos to adjust the training parameters to the dataset. For the h01 dataset we were able to build on existing experiments published by Li et al., 2020 giving us confidence in the hyperparameter selection. The comparatively fast training time of linear models trained on the embeddings (minutes) is a major strength of our method. We believe that the point of the applicability of SegCLR for supervised tasks such as subcompartment classification, especially for few samples, has been made with the extended evaluation on the h01 dataset.

Minor points:

- * The y-axes in Fig. 2d should be rescaled to better see differences.
- * Discussion of reference [32] (Huang et al., 2020) could state that SimCLR was used there as well.
- * A more recent (and presumably more relevant for EM ultrastructure) reference for attribution methods instead of [57] (Syndararajan et al., 2017) is the attribution method by Eckstein et al.:

Eckstein, Nils, et al. "Discriminative Attribution from Paired Images." Computer Vision–ECCV 2022 Workshops: Tel Aviv, Israel, October 23–27, 2022, Proceedings, Part IV. Cham: Springer Nature Switzerland, 2023.

- * Methods, below Equation 1: "512 pairs" -> "512 positive pairs"
- * Methods, "Unsupervised exploration of mouse [...]": References to Fig. 2 should be to Fig. 4
- * Methods, "Automated analysis of synaptic partners": "we λ " -> "we set λ "

Thank you! We incorporated these points into the manuscript.

Decision Letter, first revision:

Dear Viren,

Thank you for submitting your revised manuscript "Multi-Layered Maps of Neuropil with Segmentation-Guided Contrastive Learning" (NMETH-A50990A). It has now been seen by two of the original referees and their comments are below. These reviewers find that the paper has improved in revision, and therefore we'll be happy in principle to publish it in Nature Methods, pending minor revisions to satisfy the referees' final requests and to comply with our editorial and formatting guidelines.

TRANSPARENT PEER REVIEW

Nature Methods offers a transparent peer review option for new original research manuscripts submitted from 17th February 2021. We encourage increased transparency in peer review by publishing the reviewer comments, author rebuttal letters and editorial decision letters if the authors agree. Such peer review material is made available as a supplementary peer review file. Please state in the cover letter 'I wish to participate in transparent peer review' if you want to opt in, or 'I do not wish to participate in transparent peer review' if you don't. Failure to state your preference will result in delays in accepting your manuscript for publication.

ORCID

Best regards,
Nina

Nina Vogt, PhD
Senior Editor
Nature Methods

Reviewer #2 (Remarks to the Author):

The authors have addressed most of the comments and I feel like the new experiment on cross-dataset applications effectively demonstrates generalizability of the proposed method.

Although I still think that the Figure 2F requires a fully supervised baseline, given the amount of time required to train another fully supervised network, I understand the reluctance of the authors to do it.

I still feel like the authors can not make the following claim: “We used unsupervised UMAP projection to visualize samples of embeddings in the human and mouse datasets, and readily observed separate regions in UMAP space for glia versus neurons, and for axons, dendrites, and somas”. The authors should either clearly specify in the main text that they visualized a preselected manually labeled set of embeddings, or plot the available labels on the umap of a random subsample of the embeddings (enriching for the labeled ones, if necessary).

Reviewer #3 (Remarks to the Author):

The revision addresses most of the points that have been brought up. I commend especially the data and code release, which will make the method more easily available for refinement and downstream analysis. I recommend publication of the manuscript with a few minor revisions:

1. I suggest to add the response regarding the choice of the 32/40nm resolution to the discussion.
2. Figure 2d's y-axis could be rescaled a bit to make it easier to see the fine differences in F1-score.
3. Supplemental Figure 2b could show a different raw image sample to highlight that the training data is from a different dataset than in panel a.

4. Page 7, paragraph at bottom: I would appreciate if F1-scores of all alternatives would be mentioned to make it easier to assess the generalization performance (i.e., "trained on h01", "trained on MiCRONS" (with and without finetuning), and "fully supervised").

Author Rebuttal, first revision:

Reviewer #2:

The authors have addressed most of the comments and I feel like the new experiment on cross-dataset applications effectively demonstrates generalizability of the proposed method.

Although I still think that the Figure 2F requires a fully supervised baseline, given the amount of time required to train another fully supervised network, I understand the reluctance of the authors to do it.

I still feel like the authors can not make the following claim: "We used unsupervised UMAP projection to visualize samples of embeddings in the human and mouse datasets, and readily observed separate regions in UMAP space for glia versus neurons, and for axons, dendrites, and somas". The authors should either clearly specify in the main text that they visualized a preselected manually labeled set of embeddings, or plot the available labels on the umap of a random subsample of the embeddings (enriching for the labeled ones, if necessary).

*We thank the reviewer for the constructive feedback throughout the entire review process!
We clarified the highlighted section in the text as requested.*

Reviewer #3:

The revision addresses most of the points that have been brought up. I commend especially the data and code release, which will make the method more easily available for refinement and downstream analysis. I recommend publication of the manuscript with a few minor revisions:

1. I suggest to add the response regarding the choice of the 32/40nm resolution to the discussion.

Done.

2. Figure 2d's y-axis could be rescaled a bit to make it easier to see the fine differences in F1-score.

Done.

3. Supplemental Figure 2b could show a different raw image sample to highlight that the training data is from a different dataset than in panel a.

Done.

4. Page 7, paragraph at bottom: I would appreciate if F1-scores of all alternatives would be mentioned to make it easier to assess the generalization performance (i.e., "trained on h01", "trained on MICRONS" (with and without finetuning), and "fully supervised").

We added more detailed results to the text.

- Jan Funke

We thank the reviewer for the constructive feedback throughout the entire review process!

Final Decision Letter:

Dear Viren,

I am pleased to inform you that your Article, "Multi-Layered Maps of Neuropil with Segmentation-Guided Contrastive Learning", has now been accepted for publication in Nature Methods. Your paper is tentatively scheduled for publication in our December print issue, and will be published online prior to that. The received and accepted dates will be November 18th, 2022 and October 2nd, 2023. This note is intended to let you know what to expect from us over the next month or so, and to let you know where to address any further questions.

Over the next few weeks, your paper will be copyedited to ensure that it conforms to Nature Methods style. Once your paper is typeset, you will receive an email with a link to choose the appropriate publishing options for your paper and our Author Services team will be in touch regarding any additional information that may be required.

You will receive a link to your electronic proof via email with a request to make any corrections within 48 hours. If, when you receive your proof, you cannot meet this deadline, please inform us at rjsproduction@springernature.com immediately.

Please note that *Nature Methods* is a Transformative Journal (TJ). Authors may publish their research with us through the traditional subscription access route or make their paper immediately open access through payment of an article-processing charge (APC). Authors will not be required to make a final decision about access to their article until it has been accepted. [Find out more about Transformative Journals](https://www.springernature.com/gp/open-research/transformative-journals)

Authors may need to take specific actions to achieve [compliance](https://www.springernature.com/gp/open-research/funding/policy-compliance-faqs) with funder and institutional open access mandates. If your research is supported by a funder that requires immediate open access (e.g. according to [Plan S principles](https://www.springernature.com/gp/open-research/plan-s-compliance)) then you should select the gold OA route, and we will direct you to the compliant route where possible. For authors selecting the subscription publication route, the journal's standard licensing terms will need

to be accepted, including [self-archiving policies](https://www.springernature.com/gp/open-research/policies/journal-policies). Those licensing terms will supersede any other terms that the author or any third party may assert apply to any version of the manuscript.

Your paper will now be copyedited to ensure that it conforms to Nature Methods style. Once proofs are generated, they will be sent to you electronically and you will be asked to send a corrected version within 24 hours. It is extremely important that you let us know now whether you will be difficult to contact over the next month. If this is the case, we ask that you send us the contact information (email, phone and fax) of someone who will be able to check the proofs and deal with any last-minute problems.

If, when you receive your proof, you cannot meet the deadline, please inform us at rjsproduction@springernature.com immediately.

Once your manuscript is typeset and you have completed the appropriate grant of rights, you will receive a link to your electronic proof via email with a request to make any corrections within 48 hours. If, when you receive your proof, you cannot meet this deadline, please inform us at rjsproduction@springernature.com immediately.

Once your paper has been scheduled for online publication, the Nature press office will be in touch to confirm the details.

Once your paper has been scheduled for online publication, the Nature press office will be in touch to confirm the details.

Content is published online weekly on Mondays and Thursdays, and the embargo is set at 16:00 London time (GMT)/11:00 am US Eastern time (EST) on the day of publication. If you need to know the exact publication date or when the news embargo will be lifted, please contact our press office after you have submitted your proof corrections. Now is the time to inform your Public Relations or Press Office about your paper, as they might be interested in promoting its publication. This will allow them time to prepare an accurate and satisfactory press release. Include your manuscript tracking number NMETH-A50990B and the name of the journal, which they will need when they contact our office.

About one week before your paper is published online, we shall be distributing a press release to news organizations worldwide, which may include details of your work. We are happy for your institution or funding agency to prepare its own press release, but it must mention the embargo date and Nature Methods. Our Press Office will contact you closer to the time of publication, but if you or your Press Office have any inquiries in the meantime, please contact press@nature.com.

Nature Portfolio journals [encourage authors to share their step-by-step experimental protocols](https://www.nature.com/nature-research/editorial-policies/reporting-standards#protocols) on a protocol sharing platform of their choice. Nature Portfolio 's Protocol Exchange is a free-to-use and open resource for protocols; protocols deposited in Protocol Exchange are citable and can be linked from the published article. More details can found at www.nature.com/protocolexchange/about.

Best regards,
Nina

Nina Vogt, PhD
Senior Editor

Nature Methods